# A Bayesian Nonparametric Framework for Learning Disentangled Representations

**Vaishnavi S Patil**[1,4]     **Siddhi Patil**[1]     **Matthew Evanusa**[1]     **Amit Kumar Kundu**[2,4]

**Cornelia Fermüller**[3]     **Joseph JáJá**[2,3,4]

[1]Department of Computer Science     [2]Department of Electrical and Computer Engineering
[3]University of Maryland Institute for Advanced Computer Studies
[4]Institute for Health Computing
University of Maryland, College Park
{vspatil, scpatil, mevanusa, amit314, fermulcm, josephj}@umd.edu

## Abstract

Disentangled representation learning aims to identify and organize the underlying sources of variation in observed data. However, learning disentangled representations from observational data alone without any additional supervision necessitates inductive biases to solve the fundamental identifiability problem of uniquely recovering the true latent structure and parameters of the data-generating process. Existing methods address this by imposing heuristic inductive biases that typically lack these theoretical identifiability guarantees. Additionally, these methods rely on strong regularization to impose these inductive biases, creating an inherent trade-off in which stronger regularization improves disentanglement but limits the latent capacity to represent underlying variations. To address both challenges, we propose a principled generative model with a Bayesian nonparametric hierarchical mixture prior that embeds inductive biases within a provably identifiable framework for unsupervised disentanglement. Specifically, the hierarchical mixture prior imposes the structural constraints necessary for identifiability guarantees, while the nonparametric formulation allows the latent representation to scale with infinite capacity to faithfully represent the complete set of underlying variations without violating these structural constraints. To enable tractable inference under this nonparametric hierarchical prior, we develop a structured variational inference framework with a nested variational family that both preserves the hierarchical structure of the identifiable generative model and approximates the expressiveness of the nonparametric prior. We evaluate our proposed probabilistic model on standard disentanglement benchmarks, 3DShapes and MPI3D datasets characterized by diverse source variation distributions, to demonstrate that our method consistently outperforms strong baseline models through structural biases and a unified objective function, obviating the need for auxiliary regularization constraints or careful hyperparameter tuning.

## 1 Introduction

A primary objective of representation learning is not merely to perform density estimation or generate realistic samples, but to discover and characterize the latent structure inherent in observational data. This notion is formalized by disentangled representations that aim to separate the distinct, independent, and informative generative factors of variation in the data such that each variable of the representation is sensitive to changes in exactly one underlying factor while being relatively invariant to changes in others Bengio (2013). Disentangled representations have been shown to improve robustness and out-of-distribution generalization (Träuble et al., 2021; Li et al., 2024), sample efficiency in few-shot learning (Van Steenkiste et al., 2019; Cheng et al., 2024), domain adaptation via separation of transferable and domain-specific features (Tran & Huang, 2019; Cai et al., 2019), controllable and interpretable generation (Zhu et al., 2021; Wang et al., 2023; Zhou et al., 2025),

and causal inference and fairness through explicit separation of sensitive and task-relevant factors (Cheng et al., 2024; Locatello et al., 2019a).

However, unsupervised learning of disentangled representations is fundamentally challenged by identifiability which refers to whether the true generative factors and their structure can be uniquely inferred from observed data alone. Without identifiability, different parameterizations of the generative model can produce identical distributions over observed data, making it theoretically impossible to recover the true generative factors. Prior work in nonlinear independent component analysis (Hyvärinen & Pajunen, 1999; Hyvärinen et al., 2019; Khemakhem et al., 2020), deep generative modeling (Wang et al., 2021; D'Amour et al., 2022), and unsupervised disentanglement (Locatello et al., 2019b) has shown that enforcing the commonly used simple isotropic Gaussian prior in combination with a nonlinear generative function is generally insufficient to recover the true sources of variation. Without additional inductive biases, the model can learn infinitely many, potentially entangled representations that satisfy the marginal prior distribution yet fail to align with the true data-generating factors.

Moreover, prior works primarily impose heuristic inductive biases and typically rely on strong regularization to enforce them inducing an inherent trade-off whereby stronger regularization enhances disentanglement but simultaneously restricts the representation capacity. Consequently, this mis-specification of the latent capacity either under-represents all relevant modes of variation or forces encoding of the data in a manner that conflicts with the natural structure, leading to systematic violation of the disentanglement-inducing constraints.

To address both these limitations we build upon the theoretical framework of Kivva et al. (2022), who prove that mixture priors provide sufficient inductive bias for identifiability in deep generative models with piecewise affine data-generating functions. We propose a Bayesian nonparametric hierarchical mixture prior that inherits these identifiability guarantees lacking in simple Gaussian priors while simultaneously addressing the representation capacity mis-specification problem through its nonparametric formulation. The nonparametric formulation allows the complexity of factor-specific mixtures to remain unspecified a priori, a characteristic analogous to species discovery in unexplored ecosystems, where the number and types of species present cannot be predicted in advance. This formulation endows our model with universal approximation capabilities, ensuring that the identifiable architecture is, in principle, expressive enough to recover the natural underlying structure of the data. To specifically learn disentangled representations, we define a factorized prior structure under which a nonparametric hierarchical mixture prior is placed over the space of each generative factor independently, such that mixture components correspond to discrete variations of the respective factor. Consistent with the principles of classical factor analysis, this factorized structure entails that observations are generated through the combinatorial composition of factor-specific mixture components, with each observation determined by a unique combination of components across all generative factors Hsu et al. (2024a).

For tractable inference under this nonparametric hierarchical prior, we develop a structured variational inference framework with a nested variational family. The structured inference framework preserves the hierarchical structure of the identifiable generative model thereby enabling joint optimization of the prior and deep generative model parameters within a unified objective function. The nested formulation enables the variational distribution to approximate the expressiveness of the nonparametric prior while maintaining computational tractability.

Empirically, we show that this hierarchical mixture prior provides substantially stronger inductive biases enabling the learning of modular and compact disentangled representations that enhance interpretability. Our results on two image datasets with distinct factor distributions further demonstrate that the nonparametric hierarchical mixture prior and the corresponding inference framework provide sufficient inductive bias without additional computationally expensive auxiliary inductive biases or careful manual tuning of regularization hyperparameters

## 2 NONPARAMETRIC BAYESIAN PRIORS FOR AUTOENCODERS

The theoretical framework of Kivva et al. (2022) establishes identifiability of deep latent variable models by rigorously delineating the conditions under which such models are uniquely recoverable from data. Specifically, identifiability of the latent variable distribution up to affine transformations

requires two critical conditions: (i) the marginal distribution of the latent variables follows a Gaussian mixture model (GMM), and (ii) the corresponding data-generating function is piecewise affine and satisfies weak injectivity. Building on these conditions, the authors further demonstrate that stronger identifiability, up to permutation, scaling, and translation, necessitates the introduction of additional discrete latent variables which index the mixture components of the GMM, inducing a hierarchical structure in which the continuous latent variables are conditionally independent given these discrete variables. Furthermore, imposing an additional maximality condition (P3 in Kivva et al. (2022)) on this hierarchical structure yields identifiability of the complete latent representation, including the latent dimensionality, the cardinality and distribution of the discrete variables, and the conditional distribution of the continuous latent variables.

To further bias this identifiable framework toward learning disentangled representations, we impose an additional structural constraint on the discrete latent space. Specifically, we factorize the multivariate discrete latent variable into statistically independent components, where each component encodes a distinct factor of variation and indexes a set of mixture components representing the discrete modes of variations of its respective factor. Critically, this factorized prior enforces an orthogonal partitioning of the latent space, such that factor-specific variations are encoded in non-overlapping subspaces of the continuous latent representation, thereby promoting the independent encoding of factor-specific variations. Moreover, this factorized latent space induces a generative process in which observations are generated through the combinatorial composition of factor-specific mixture components, with each observation generated by a unique combination of component assignments.

This inductive bias of combinatorial composition toward learning disentangled representations is closely related to the works of Hsu et al. (2024a;b), which similarly structure the latent space as a product of independent discrete sets to promote disentangled representations. In their framework, the discrete latent space is formed from continuous latent representations that are discretized via vector quantization, wherein each latent dimension is independently mapped to discrete codes through distinct, learnable codebooks. For this architecture to serve as an effective inductive bias toward disentanglement, the size of each codebook, and thus the cardinality of the corresponding discrete latent variable indexing its embeddings, is kept small and fixed, constraining each dimension to capture only a limited set of discrete modes. While this design choice encourages parsimonious representations, it imposes a rigid upper bound on the expressive capacity of each factor: factors whose discrete modes of variations exceed the cardinality of a single codebook must either be distributed across multiple independent codebooks or be only partially captured, in both cases reducing the interpretability and semantic coherence of the learned latent factors.

To address the expressive limitations of fixed-cardinality discrete variables, we introduce a nonparametric Dirichlet Process mixture (Ferguson, 1973; Sethuraman, 1994; Antoniak, 1974) (see Section A.1.1 for a detailed description) prior over the latent space, allowing the support of each latent variable to grow adaptively with the intrinsic complexity of its associated factor of variation. By placing nonzero probability mass over a countably infinite support, the DP prior equips each discrete latent variable with a universal approximation property over discrete distributions, enabling it to represent an arbitrary discrete distribution over the modes of its associated factor. Consequently, this universal approximation property reinforces the statistical independence of the joint latent prior, ensuring that the complete set of discrete modes of variation associated with each independent factor is represented entirely by a discrete latent variable. Critically, because the DP prior is applied independently to each discrete latent variable, each factor can adaptively expand its variation space without disrupting the orthogonal partition structure induced by the remaining factors, thereby preserving statistical independence of the representation.

In the following sections, we formalize this framework by deriving a Bayesian nonparametric hierarchical generative model that is jointly identifiable, universally expressive through the nonparametric prior, and structurally biased toward disentangled representations with a particular emphasis on completeness and interpretability. We then formalize the structured variational family, which is specifically designed to accurately approximate the posterior distribution under the hierarchical factorized structure of the prior. Finally, we present a nested extension of this structured variational family that enables principled incremental expansion of representational capacity, with the structured component preserving the inductive biases of identifiability and factorial independence, and the nested component approximating the universal approximation capacity of the nonparametric prior, jointly ensuring that the learning algorithm remains faithful to the theoretical properties of the generative model throughout inference.

## 2.1 GENERATIVE MODEL WITH A NONPARAMETRIC PRIOR

We consider a hierarchical latent variable generative model in which each observation $x \in \mathcal{X}$ is generated via a structured latent representation comprising a continuous latent variable $e \in \mathbb{R}^d$ and a multivariate discrete latent variable $z = (z_1, \ldots, z_d) \in \mathcal{Z}$ with $\mathcal{Z}$ denoting the corresponding discrete latent space. To impose an inductive bias toward disentangled representations, we assume a factorial structure on the discrete space $\mathcal{Z}$, defined as the Cartesian product $\mathcal{Z} = \mathcal{Z}_1 \times \cdots \times \mathcal{Z}_d$, where each $z_i \in \mathcal{Z}_i = \{1, \ldots\}$ is mutually independent and captures a distinct factor of variation. To ensure identifiability of the continuous latent variable $e$ up to affine transformations, we follow the conditions of Kivva et al. (2022) and specify the marginal distribution of $e$ as a Gaussian mixture model (Assumption P1). To further satisfy the conditions for stronger identifiability (Assumption P2), we require that, conditioned on the discrete latent variable $z$, the components of $e$ are mutually independent, $e_i \perp\!\!\!\perp e_j \mid z$ for all $i \neq j$ enforcing an orthogonal partitioning of the continuous latent space into $d$ non-overlapping factor-specific subspaces.

To allow an unbounded number of mixture components, we model each $e_i$ under an independent Dirichlet Process mixture prior, inducing a countably infinite Gaussian mixture over the marginal $p(e_i)$. Formally, we place a DP prior $G_i \sim \mathrm{DP}(\alpha, G_0)$ over the component parameters, where $\alpha > 0$ is the concentration parameter controlling the richness of the induced partition, and $G_0$ is the base distribution over the parameter space $\Theta_i$. Under the stick-breaking representation of the Dirichlet Process, we define an infinite sequence of stick breaking proportions $\beta_i = \{\beta_{i,k}\}_{k=1}^{\infty}$, with $\beta_{i,k} \sim \mathrm{Beta}(1, \alpha)$ and draw the corresponding component parameters independently from the base distribution, $\theta_{i,k} \sim G_0(\lambda)$ as follows,

$$\beta_{i,k} \mid \alpha \sim p(\beta \mid \alpha) = \mathrm{Beta}(1, \alpha), \quad \theta_{i,k} \mid \lambda \sim p(\theta \mid \lambda) = G_0(\lambda), \quad \forall i \in \{1, \ldots d\}, \forall k \in \mathbb{N}$$

From the stick-breaking proportions $\beta_i$, we construct an infinite sequence of mixing weights $\pi_{i,k} = \beta_{i,k} \prod_{j=1}^{k-1}(1 - \beta_{i,j})$, satisfying $\sum_{k=1}^{\infty} \pi_{i,k} = 1$ almost surely, yielding the random discrete measure $G_i = \sum_{k=1}^{\infty} \pi_{i,k} \delta_{\theta_{i,k}}$. The discrete latent variable $z_i$ then selects a component index according to $z_i \sim \mathrm{Categorical}(\pi_i)$, upon which the continuous component $e_i$ is drawn from the corresponding component-specific distribution parameterized by $\theta_{i,k} \in \Theta_i$, i.e. $e_i \mid z_i = k, \theta_i \sim p(e_i \mid \theta_{i,k})$.

$$z_i = k \mid \beta_i \sim p(z_i = k \mid \beta_i) = \pi_{i,k}, \quad e_i \sim p(e \mid z_i, \theta_i) = \prod_{k=1}^{\infty} \left( p(e \mid \theta_{i,k}) \right)^{\mathbf{1}_{[z_i = k]}}$$

We specify each component-specific distribution as a Gaussian with unknown mean and precision, $p(e_i \mid \theta_{i,k}) = \mathcal{N}(e_i \mid \mu_{i,k}, s_{i,k}^{-1})$, where $\theta_{i,k} = (\mu_{i,k}, s_{i,k})$ denotes the mean and precision parameters of the $k$-th component. Exploiting conjugacy, we set the base distribution $G_0$ to be a Normal-Gamma distribution, $G_0(\lambda) = \mathcal{NG}(m_0, \kappa_0, \nu_0, w_0)$, which serves as the natural conjugate prior to the Gaussian likelihood with unknown mean and precision, enabling closed-form posterior updates over $\theta_{i,k}$. Further, to enforce the maximality condition, Assumption P3 of Kivva et al. (2022), we place a Gamma prior on $\alpha$, biasing the model toward the minimal number of active discrete components consistent with the observed data, thereby selecting a unique representative within the equivalence class of marginal-preserving representations.

Completing the generative hierarchy, the observation $x$ is then generated via a piecewise affine deep neural network (with Rectified Linear Unit (ReLU) nonlinearities) data-generating function $g_{\theta_g} : \mathbb{R}^d \to \mathcal{X}$ satisfying the weak injectivity conditions (F1 and F2 of Kivva et al. (2022)), with additive observation noise: $x \mid e \sim p_{\theta_g}(x \mid e) = \mathcal{P}(g_{\theta_g}(e))$ where $\mathcal{P}$ denotes an appropriate likelihood family.

Together, these modeling choices yield a generative model with three key properties: identifiability of the latent space up to permutation, scaling, and translation; nonparametric expressiveness through the DP prior; and an explicit factorial structure promoting disentangled, factor-specific representations. The joint distribution over observations $x$, continuous variables $e$, discrete variables $z$, stick-breaking proportions $\beta$, and component parameters $\theta$ factorizes according to the hierarchical generative process as

$$p(x, e, z, \beta, \theta \mid \alpha, \lambda) = p_{\theta_g}(x \mid e) \prod_{i=1}^{d} p(e_i \mid z_i, \theta_i) \, p(z_i \mid \beta_i) \prod_{k=1}^{\infty} p(\beta_{i,k} \mid \alpha) \, p(\theta_{i,k} \mid \lambda) \quad (1)$$

where the outer product over $i$ encodes the factorial independence across latent dimensions, and the inner product over $k$ reflects the infinite mixture structure induced by the Dirichlet Process prior.

## 2.2 Structured Nested Variational Family

To enable tractable posterior approximation under the nonparametric DP prior (see Section A.1 for a primer on variational inference for DPMMs), existing methods Blei & Jordan (2006); Hoffman et al. (2013) rely on two simplifying assumptions. First, they approximate the infinite-dimensional stick-breaking process using a truncated variational family by introducing an explicit truncation level $T$, fixing $q_{\nu_{\beta_i}}(\beta_{i,T} = 1) = 1$ for each dimension $i$, thereby forcing all subsequent stick-breaking weights $\{\beta_{i,k}\}_{k>T}$ and their corresponding mixing weights to zero, limiting the mixture components to $T$. Second, they adopt a mean-field variational family, imposing full factorization across all latent variables, thereby severing the hierarchical dependencies induced by the generative model.

We address the second limitation by adopting the structured variational framework of Hoffman & Blei (2015), which relaxes the mean-field assumption and retains the conditional dependencies induced by the generative model, yielding more accurate posterior approximations with reduced bias and lower sensitivity to initialization and hyperparameters. Specifically, for each dimension $i$, we preserve the hierarchical dependencies between the stick-breaking proportions $\boldsymbol{\beta}_i$ and the discrete variable $z_i$ along with the component parameters $\boldsymbol{\theta}_i$, and the continuous latents $e_i$ with the following structured variational family,

$$q_\nu(e_i, z_i, \boldsymbol{\beta}_i, \boldsymbol{\theta}_i) = q(e_i \mid z_i, \boldsymbol{\theta}_i) q(z_i \mid \boldsymbol{\beta}_i) \prod_{k=1}^{T-1} q_{\nu_{\beta_{i,k}}}(\beta_{i,k}) \prod_{k=1}^{T} q_{\nu_{\theta_{i,k}}}(\theta_{i,k}) \qquad (2)$$

Truncated variational families render inference in infinite-mixture models tractable by restricting the variational distribution to a fixed number $T$ of mixture components. Intuitively, one might expect the ELBO to increase monotonically with $T$, as additional components allow the variational posterior to better approximate the true nonparametric posterior, thereby motivating the use of large truncation levels. However, when the data are generated by a finite mixture, there can exist an optimal truncation level beyond which additional components do not improve the approximation, because the truncated families are not nested: the variational family with truncation level $T$ is not a subset of the family with truncation level $T + 1$. Consequently, increasing $T$ beyond the optimal level does not necessarily improve the approximation, and crucially, undermines the inductive bias toward parsimonious representations Hsu et al. (2024a) that is essential for learning disentangled factors.

To address this, we employ the nested variational family of Kurihara et al. (2006), which supports an unbounded number of components in the variational family by tying the variational parameters of all components beyond an implicit truncation level $T$ to their prior values:

$$q_{\nu_\beta}(\boldsymbol{\beta}_i) = \prod_{k=1}^{T} q_{\nu_{\beta_{i,k}}}(\beta_{i,k}) \prod_{k=T}^{\infty} p(\beta_{i,k} \mid \alpha), \qquad q_{\nu_\theta}(\boldsymbol{\theta}_i) = \prod_{k=1}^{T} q_{\nu_{\theta_{i,k}}}(\theta_{i,k}) \prod_{k=T}^{\infty} p(\theta_{i,k} \mid \lambda) \quad (3)$$

Under this formulation, only the first $T$ components have free variational parameters, ensuring that despite the variational distribution supporting an unbounded number of components only $T$ sets of parameters need to be represented and optimized, rendering inference computationally tractable. Crucially, under this nested variational family, data points that are not explained by the first $T$ components with free variational parameters can still be assigned to components beyond $T$, whose parameters remain tied to the prior. This enables a greedy inference procedure that begins with $T = 1$ and incrementally adds components only when they yield a significant improvement in the ELBO, continuing until all data are assigned to components within the active truncation level and no mass remains on prior-tied components. Notably, under this nested formulation, components that fail to encode meaningful variations collapse back to their prior distribution during training.

Given the generative model in equation 1 and the hierarchical structured nested variational family $q_\nu$, the evidence lower bound (ELBO) for a single data point, estimated via Monte Carlo, can be written as

$$\mathcal{L} = \mathbb{E}_{q_{\nu_\beta}} \left[ \log \frac{p(\boldsymbol{\beta} \mid \alpha)}{q_{\nu_\beta}(\boldsymbol{\beta})} \right] + \mathbb{E}_{q_{\nu_\theta}} \left[ \log \frac{p(\boldsymbol{\theta} \mid \lambda)}{q_{\nu_\theta}(\boldsymbol{\theta})} \right] + \mathbb{E}_{q_\nu} \left[ \log \frac{p_{\theta_g}(x \mid e) \, p(e \mid z, \boldsymbol{\theta}) \, p(z \mid \boldsymbol{\beta})}{q(e \mid z, \boldsymbol{\theta}) \, q(z \mid \boldsymbol{\beta})} \right]$$

## 2.3 THE INFERENCE ALGORITHM

Efficient inference in hierarchical models relies on conjugate exponential family likelihoods, which preserve tractable conditional posterior distributions within the same exponential family as the prior admitting closed-form updates. For general neural network observation models such as $p_{\theta_g}$, the absence of conjugacy makes inference over latent variables substantially more expensive and often requires multiple passes through the generative network for a single update. To retain the computational advantages of conjugate structured inference while using flexible non-conjugate likelihoods, we follow Johnson et al. (2016) and use deep amortized recognition networks $h(x; \phi)$ that output local conjugate likelihood potentials $\hat{p}_\phi$ (Eq. 4), rather than directly parameterizing the variational distributions as in standard variational autoencoders. These conjugate potentials serve as amortized, data-dependent surrogates for the intractable likelihood terms during inference and are combined with the structured latent prior via message passing, leaving the generative model itself unchanged.

$$\hat{p}_\phi(\boldsymbol{e} \mid x) = \prod_{i=1}^d \hat{p}_\phi(e_i \mid x) = \prod_{i=1}^d \exp\{\langle h_i(x; \phi), t_e(e_i)\rangle\} \tag{4}$$

This factored structure of the recognition network mirrors the factorial prior over $e_i$, allowing inference for the pair of local latent variables $(e_i, z_i)$ associated with each dimension $i$ to be inferred independently. Consequently, the data-dependent term of the ELBO in Eq. 2.2 decomposes into a sum of local contributions $\mathcal{L}_i$, one per dimension:

$$\mathcal{L}_i = \mathbb{E}_{q_{\nu_\beta}(\boldsymbol{\beta}_i)q(z_i|\boldsymbol{\beta}_i)} \left[\log \frac{p(z_i \mid \boldsymbol{\beta}_i)}{q(z_i \mid \boldsymbol{\beta}_i)} + \mathbb{E}_{q_{\nu_\theta}(\boldsymbol{\theta}_i)q(e_i|z_i,\boldsymbol{\theta}_i)} \left[\log \frac{p(e_i \mid z_i, \boldsymbol{\theta}_i)}{q(e_i \mid z_i, \boldsymbol{\theta}_i)} + \log \hat{p}_\phi(e_i \mid x)\right]\right] \tag{5}$$

Following Hoffman & Blei (2015), conditioning on the global variables $\boldsymbol{\beta}_i$ and $\boldsymbol{\theta}_i$, for any given value of the local variable $z_i = k$, the inner expectation over $e_i$ admits the form of a variational lower bound on the conditional marginal likelihood of the conjugate potential:

$$\mathbb{E}_{q(e_i|z_i,\theta_i)} \left[\log \frac{p(e_i \mid z_i, \boldsymbol{\theta}_i)}{q(e_i \mid z_i, \boldsymbol{\theta}_i)} + \hat{p}_\phi(e_i \mid x)\right]$$
$$= -D_{\mathrm{KL}}\left(q(e_i \mid z_i, \boldsymbol{\theta}_i)\|\hat{p}_\phi(e_i \mid x_i, z_i, \boldsymbol{\theta}_i)\right) + \log \hat{p}_\phi(x_i \mid z_i, \boldsymbol{\theta}_i) \le \log \hat{p}_\phi(x_i \mid z_i, \boldsymbol{\theta}_i) \tag{6}$$

where the local posterior and conjugate marginal likelihood are defined as

$$\hat{p}_\phi(e_i \mid x_i, z_i, \boldsymbol{\theta}_i) = \frac{\hat{p}_\phi(e_i \mid x_i)p(e_i \mid z_i, \boldsymbol{\theta}_i)}{\hat{p}_\phi(x_i \mid z_i, \boldsymbol{\theta}_i)}, \ \hat{p}_\phi(x_i \mid z_i, \boldsymbol{\theta}_i) = \int \hat{p}_\phi(e_i \mid x_i) \, p(e_i \mid z_i, \boldsymbol{\theta}_i) de_i \tag{7}$$

Since the Kullback-Leibler (KL) divergence is non-negative, setting $q(e_i \mid z_i, \boldsymbol{\theta})$ equal to the local posterior $\hat{p}_\phi(e_i \mid x_i, z_i, \boldsymbol{\theta})$ minimizes the KL divergence to zero and yields the tightest lower bound on the local ELBO. Under conjugacy, this optimal variational distribution takes the exponential family form:

$$q(e_i \mid z_i, \boldsymbol{\theta}_i) = \hat{p}_\phi(e_i \mid x_i, z_i, \boldsymbol{\theta}_i)$$
$$= \exp\left\{\langle \eta_e(z_i, \eta_\theta(\boldsymbol{\theta}_i), \phi), t_e(e)\rangle - A_e(\eta_e(z_i, \eta_\theta(\boldsymbol{\theta}_i), \phi))\right\}$$
$$\eta_e(z_i, \eta_\theta(\boldsymbol{\theta}_i), \phi) = \sum_{k=1}^T \mathbf{1}_{[z_i=k]} \, \eta_\theta(\theta_{i,k}) + h_i(x_i; \phi) \tag{8}$$

with natural parameters which explicitly combines the structured prior through $\eta_\theta(\theta_{i,k})$ with the data-dependent signal from the recognition network $h_i(x; \phi)$. With this optimal choice of $q(e_i \mid z_i, \boldsymbol{\theta}_i)$, the local ELBO reduces to a variational lower bound on the marginal likelihood of the data conditioned on the global variable $\boldsymbol{\beta}_i$ as follows:

$$\mathcal{L}_i = \mathbb{E}_{q_{\nu_\beta}(\boldsymbol{\beta}_i)q(z_i|\boldsymbol{\beta}_i)} \left[\log \frac{p(z_i \mid \boldsymbol{\beta}_i)}{q(z_i \mid \boldsymbol{\beta}_i)} + \mathbb{E}_{q_{\nu_\theta}(\boldsymbol{\theta}_i)} \left[\log \hat{p}_\phi(x \mid z_i, \boldsymbol{\theta}_i)\right]\right] \tag{9}$$
$$= -D_{\mathrm{KL}}(q(z_i \mid \boldsymbol{\beta}_i)\|\hat{p}_\phi(z_i \mid x_i, \boldsymbol{\beta}_i)) + \log \hat{p}_\phi(x_i \mid \boldsymbol{\beta}_i) \le \log \hat{p}_\phi(x_i \mid \boldsymbol{\beta}_i)$$

and similar to the variational distribution of $e_i$, the locally optimal variational distribution over $z_i$ for $k \le T$ is given by:

$$\log q(z_i = k \mid \boldsymbol{\beta}_i) \propto \log p(z_i = k \mid \boldsymbol{\beta}_i) + \mathbb{E}_{q_{\nu_\theta}(\boldsymbol{\theta}_i)} \log \hat{p}_\phi(x_i \mid \theta_{i,k})$$
$$= \log \pi_{i,k} + \mathbb{E}_{q_{\nu_\theta}(\boldsymbol{\theta}_{i,k})}[A_e(\eta_\theta(\theta_{i,k}) + h_i(x_i; \phi)) - A_e(\eta_\theta(\theta_{i,k}))] \tag{10}$$

While the nested variational family supports an unbounded number of components, naively computing the variational assignment $q(z_i = k \mid \boldsymbol{\beta}_i)$ for all $k > T$ would require sampling infinitely many stick-breaking proportions to evaluate the corresponding mixing weights $\pi_{i,k}$. However, two properties of the nested family together enable a tractable closed-form estimate. First, since the parameters of all components $k > T$ are tied to the prior $p(\boldsymbol{\theta}_{i,k} \mid \lambda)$, the expected log-likelihood $\mathbb{E}_{p(\theta_{i,k} \mid \lambda)}[\log \hat{p}_\phi(x_i \mid \theta_{i,k})]$ is identical across all components beyond $T$ and can be factored out of the infinite sum. Second, since the mixing weights are normalized and satisfy $\sum_{k=1}^\infty \pi_{i,k} = 1$ almost surely, the total weight assigned to all components beyond $T$ reduces to $(1 - \sum_{k=1}^T \pi_{i,k})$, which depends only on the $T$ free variational parameters. Together, these two properties yield the following tractable expression for the total assignment probability to all prior-tied components:

$$
\begin{aligned}
\log q(z_i = k \mid \boldsymbol{\beta}_i) &\propto \log p(z_i = k \mid \boldsymbol{\beta}_i) + \mathbb{E}_{p(\theta_{i,k} \mid \lambda)} \log \hat{p}_\phi(x_i \mid \theta_{i,k}) \\
q(z_i = k \mid \boldsymbol{\beta}_i) &\propto \pi_{i,k} \cdot \exp\Big\{ \mathbb{E}_{p(\theta \mid \lambda)} \log \hat{p}_\phi(x_i \mid \theta) \Big\} \\
q(z_i > T \mid \boldsymbol{\beta}_i) &= \sum_{k=T+1}^\infty q(z_i = k \mid \boldsymbol{\beta}_i) \\
&= \sum_{k=T+1}^\infty \pi_{i,k} \cdot \exp\Big\{ \mathbb{E}_{p(\theta \mid \lambda)} \log \hat{p}_\phi(x_i \mid \theta) \Big\} \\
&= \Big( 1 - \sum_{k=1}^T \pi_{i,k} \Big) \cdot \exp\Big\{ \mathbb{E}_{p(\theta \mid \lambda)} \log \hat{p}_\phi(x_i \mid \theta) \Big\}
\end{aligned}
\tag{11}
$$

This quantity serves as the stopping criterion for the greedy component-expansion procedure where a new component is added to the active truncation level when $q(z_i > T \mid \boldsymbol{\beta}_i)$ exceeds a threshold, indicating that residual data cannot be explained by the current $T$ components.

As in structured variational inference more broadly, restoring dependencies between global variables and local variables along with amortized recognition networks enables the local variational distributions to be set to their locally optimal values in closed form. At each iteration, the global parameters $\boldsymbol{\beta}_i$ and $\boldsymbol{\theta}_i$ are sampled from $q_{\nu_\beta}(\boldsymbol{\beta}_i)$ and $q_{\nu_\theta}(\boldsymbol{\theta}_i)$ respectively using the reparameterization trick, and used to compute the local variational distributions. Since exact inference over the global variables remains intractable due to the hierarchical dependencies (Hoffman & Blei, 2015), and since the use of recognition networks precludes analytic gradient computation, we optimize the ELBO using low-variance Monte Carlo estimators, yielding unbiased gradient estimates for both global and recognition network parameters. Step sizes are selected following the principled stability and convergence guidelines of Mandt et al. (2017), with optimization carried out using adaptive preconditioned gradient methods, specifically Adam (Kingma & Ba, 2017).

## 3 EXPERIMENTS

In this section, we present experiments designed to empirically assess whether the hierarchical Bayesian nonparametric approach to latent quantization provides effective inductive biases for learning interpretable disentangled representations. Specifically, we evaluate whether our approach achieves comparable performance relative to prior work which impose equivalent inductive biases through multiple, distinct regularization terms.

**Datasets.** Our experimental framework systematically addresses these questions through comprehensive quantitative evaluations conducted on two benchmark datasets labeled with ground-truth source information. Each dataset is constructed from mutually independent sources through a deterministic data generation process. In particular, we use the 3DShapes (Burgess & Kim, 2018) dataset of 3D shapes generated from six ground-truth independent latent factors with approximately uniform and small number of variations. Additionally, we use the MPI3D dataset (Gondal et al., 2019), collected from a real-world robotic environment, which exhibits a power-law distribution across the number of variations of different factors. Specifically, a few factors contain extensive variations (e.g., 40 discrete values for each rotational degrees of freedom), while the majority possess substantially fewer variations (e.g., 2-6 values for object properties).

**Prior Methods.** We evaluate our proposed approach against several state-of-the-art methods that incorporate distinct inductive biases for unsupervised disentanglement. Specifically, we compare to $\beta$-VAE Higgins et al. (2017) and $\beta$-TCVAE Chen et al. (2018) which enforce disentanglement through information-theoretic regularization encouraging independence across latent dimensions. We further consider BioAE Whittington et al. (2022), which introduces biologically inspired constraints—namely nonnegativity and energy efficiency—to promote compact representations enforcing neurons to become selective for single factors of task variation, together with a grid-like structural constraint as an architectural inductive bias. In addition, we examine QLAE Hsu et al. (2024a), which introduces an architectural bias based on latent quantization, and subsequently Tripod Hsu et al. (2024a), which combines latent quantization with additional inductive biases enforcing independence among latent variables as well as constraining the functional mapping from latent representations to the data space. For a concise primer on the disentanglement metrics and the different properties measured please refer to Section A.4.

**Quantitative Comparison with Prior Methods.** The experimental evaluation demonstrates that the proposed Bayes-QLAE consistently outperforms most baseline methods across both datasets in terms of modularity metrics (InfoM and D), with the notable exception of achieving competitive performance relative to QLAE and Tripod (Table 1 and Table 2). The observed improvements in compactness (InfoC) are particularly pronounced when compared to the baseline QLAE, demonstrating the effectiveness of the nonparametric prior in adapting to the complexity of underlying generative factors while maintaining consistency in modularity. Contrary to the position advanced by the authors of QLAE, who prioritize modularity/disentanglement over compactness/completeness through specific architectural design choices, we argue that achieving interpretable representations that faithfully capture mutually independent generative factors requires balanced weighting of both modularity and compactness metrics. With competitive explicitness and informativeness measures, Bayes-QLAE demonstrates performance consistent with QLAE and Tripod while substantially outperforming alternative approaches, reinforcing the efficacy of latent quantization for disentangled representation learning.

It is worth noting that Tripod achieves its superior modularity and compactness performance through the application of a Normalized Hessian Penalty, which necessitates multiple forward passes through the generative network, thereby incurring additional computational overhead. In contrast, Bayes-QLAE achieves competitive performance through architectural inductive biases alone, without requiring additional regularization terms. Furthermore, Tripod's disentanglement performance, particularly in modularity and compactness dimensions, exhibits sensitivity to quantization level hyperparameters, which must be specified a priori. Conversely, Bayes-QLAE demonstrates adaptive behavior that automatically learns quantization levels from the data while maintaining robustness across evaluation metrics.

We observe that the performance improvement of Bayes-QLAE is notably more pronounced on the 3DShapes dataset compared to the MPI3D dataset, where factor variations are characterized by a power-law distribution. This differential performance suggests that the underlying distributional properties of the generative factors significantly influence the efficacy of the nonparametric prior. We hypothesize that replacing the Dirichlet Process prior with a more flexible, generalized prior such as the Pitman–Yor process—which allows for a richer clustering structure and can model power-law behaviors—may yield further performance gains. We perform detailed ablation studies to systematically isolate and quantify the inductive biases contributed by each component of our hierarchical Bayesian nonparametric framework in Section A.3.

**Qualitative Comparisons**. We conduct qualitative assessments of our proposed method on each dataset to evaluate both sample reconstructions and latent traversals in Section A.5. Furthermore, we also assess the evolution of the latent traversals for the 3DShapes dataset with an increase in the number of components in Figure 2.

## 4 RELATED WORKS

The challenge of separating mutually independent sources in data traces back to the classical statistical problem of Independent Component Analysis (ICA) Comon (1994); Hyvärinen & Oja (2000). This core problem was later reinterpreted in the context of modern machine learning as disentanglement, formally articulated by Bengio Bengio (2013) and formalized by Higgins et al. (2018). When

| model | InfoM | InfoC | InfoE | D | C | I |
|-------|-------|-------|-------|---|---|---|
| $\beta$-VAE | $0.62 \pm .02$ | $0.44 \pm .03$ | $0.93 \pm .02$ | $0.58 \pm .02$ | $0.42 \pm .02$ | $\mathbf{0.97} \pm .02$ |
| $\beta$-TCVAE | $0.65 \pm .03$ | $\mathbf{0.56} \pm .02$ | $0.91 \pm .02$ | $0.56 \pm .02$ | $0.46 \pm .02$ | $\mathbf{0.95} \pm .02$ |
| BioAE | $0.58 \pm .02$ | $0.42 \pm .02$ | $0.90 \pm .01$ | $0.48 \pm .01$ | $0.39 \pm .02$ | $0.91 \pm .02$ |
| QLAE | $0.84 \pm .02$ | $0.49 \pm .01$ | $\mathbf{0.97} \pm .01$ | $\mathbf{0.79} \pm .01$ | $0.56 \pm .01$ | $\mathbf{0.97} \pm .01$ |
| Tripod | $\mathbf{0.91} \pm .03$ | $\mathbf{0.58} \pm .03$ | $0.96 \pm .02$ | $0.80 \pm .03$ | $0.63 \pm .03$ | $\mathbf{0.97} \pm .02$ |
| Bayes-QLAE | $\mathbf{0.91} \pm .03$ | $\mathbf{0.61} \pm .02$ | $0.95 \pm .02$ | $\mathbf{0.84} \pm .03$ | $\mathbf{0.65} \pm .03$ | $\mathbf{0.97} \pm .02$ |

Table 1: Disentanglement metrics measured in InfoMEC and DCI for 3DShapes dataset. For each metric a higher score is better. The scores for all the models were averaged across 5 runs with different random seeds with intervals denoting 95% confidence intervals of the mean estimated assuming a t-distribution. The results for the VQE-based and QLAE-based models are obtained using the hyperparameter settings and experimental conditions as described in Locatello et al. (2019b) and Hsu et al. (2024a;b) respectively.

| model | InfoM | InfoC | InfoE | D | C | I |
|-------|-------|-------|-------|---|---|---|
| $\beta$-VAE | $0.41 \pm .03$ | $0.40 \pm .03$ | $\mathbf{0.68} \pm .03$ | $0.24 \pm .03$ | $0.19 \pm .03$ | $\mathbf{0.80} \pm .03$ |
| $\beta$-TCVAE | $0.48 \pm .03$ | $0.46 \pm .03$ | $0.62 \pm .03$ | $0.27 \pm .03$ | $0.24 \pm .03$ | $\mathbf{0.79} \pm .03$ |
| BioAE | $0.44 \pm .03$ | $0.38 \pm .02$ | $0.61 \pm .03$ | $0.26 \pm .02$ | $0.14 \pm .02$ | $0.77 \pm .02$ |
| QLAE | $0.52 \pm .02$ | $0.43 \pm .02$ | $\mathbf{0.68} \pm .04$ | $0.38 \pm .04$ | $0.34 \pm .04$ | $0.81 \pm .04$ |
| Tripod | $\mathbf{0.59} \pm .05$ | $\mathbf{0.54} \pm .05$ | $\mathbf{0.74} \pm .06$ | $0.47 \pm .04$ | $0.45 \pm .05$ | $\mathbf{0.84} \pm .05$ |
| Bayes-QLAE | $\mathbf{0.60} \pm .03$ | $\mathbf{0.56} \pm .03$ | $0.71 \pm .04$ | $\mathbf{0.48} \pm .03$ | $\mathbf{0.47} \pm .03$ | $0.81 \pm .03$ |

Table 2: Disentanglement metrics measured in InfoMEC and DCI for MPI3D dataset. For each metric a higher score is better. The scores for all the models were averaged across 5 runs with different random seeds with intervals denoting 95% confidence intervals of the mean estimated assuming a t-distribution.

the data generating process is governed by nonlinear transformations Hyvärinen & Pajunen (1999), the task of learning disentangled representations becomes theoretically unidentifiable Hyvärinen & Oja (2000); Khemakhem et al. (2020); Locatello et al. (2019b). Consequently, the incorporation of auxiliary data Hyvärinen & Pajunen (1999); Hyvärinen et al. (2019); Khemakhem et al. (2020) or weak supervision Shu et al. (2019); Locatello et al. (2020) is necessary to achieve identifiability in disentanglement. A distinct line of research focuses on the incorporation of inductive biases either in the model, training objective, or the data (Locatello et al., 2019b) for identifiability.

**Information-theoretic Regularization Biases**. Many early and influential works leverage information-theoretic constraints on the latent space to encourage factorization. The $\beta$-VAE variants Higgins et al. (2017); Burgess et al. (2018) introduces a scalar multiplicative factor on the KL divergence penalty with isotropic Gaussian priors, forming an information bottleneck, limiting the amount of information each latent can capture. Extensions like FactorVAE Kim & Mnih (2018) and $\beta$-TCVAE Chen et al. (2018) further refine these constraints by explicitly penalizing total correlation to enforce statistical independence between dimensions. BioAE (Whittington et al., 2022) demonstrates that biologically inspired constraints, specifically, minimizing latent activity and weight energy while promoting latent non-negativity encourage more factorized representations. In a similar vein, temporal sparsity is used to encourage the learning of factors varying independently across sequences (Sprekeler et al., 2014; Klindt et al., 2020).

**Architectural and Structural Biases**. Structural inductive biases embedded directly into model architectures have proved powerful. Vector quantization in models like QLAE (Hsu et al., 2024a) and the recent Tripod framework (Hsu et al., 2024b) induce grid-like latent spaces that simplify factor separation. FactorQLAE (Baykal et al., 2024) combine scalar quantization of the latent variables with a total correlation term in the optimization as an inductive bias. On the theoretical front, Barin-Pacela et al. (2024) establish identifiability for quantized factors under nonlinear mappings. Further, Leeb et al. (2020) demonstrate that restricting different latents to enter the decoding computation graph at different points can enable disentanglement. Diffusion-based architectural biases have emerged as particularly effective inductive structures. Yang et al. (2023) introduce the first un-

supervised framework for disentangling pre-trained diffusion models by automatically discovering latent factors and decomposing gradient fields into factor-conditioned sub-gradients. Further, Yang et al. (2024) show that diffusion models with cross-attention mechanisms serve as strong inductive biases, relying on the inherent information bottlenecks in the diffusion process and cross-attention mechanisms. Dynamic Gaussian Anchoring in Jun et al. (2025) bias towards a cluster structure in the latent space of diffusion models with cross-attention mechanisms for better separability between factor variations. Compositional constraints offer another structural approach, where maximizing the validity of composite images generated through stochastic mixing operators between latent representations enforces meaningful factor recombination without factor-specific architectural biases (Jung et al., 2025).

Recent work emphasizes the incorporation of multiple, complementary inductive biases, for example, Tripod integrates quantization, statistical independence, and inter-latent influence minimization into a single framework Hsu et al. (2024b). Similarly, our work combines complementary inductive biases derived from nonparametric priors, structured variational inference, and stochastic quantization, within a principled Bayesian framework with a unified objective that provides theoretical grounding for their integration.

## 5 CONCLUSION

In this paper, we introduce a novel approach that incorporates Bayesian nonparametric priors into the latent space of autoencoders. By leveraging the flexibility of nonparametric Bayesian methods, our approach enables the model to adaptively partition the latent space in accordance with the underlying data complexity, promoting more interpretable and structured latent encodings. This prior biases the learned representations toward capturing the underlying structure inherent in the data, thereby facilitating the learning of disentangled representations.

To enable accurate posterior inference under this flexible and hierarchical prior, we introduce a tailored nested and structured variational family. This variational family is specifically designed to preserve both the hierarchical structure of the prior and the inductive bias imposed by a discrete latent space, ensuring that the inference procedure remains expressive enough to capture complex dependencies while maintaining the structural properties essential for effective representation learning.

Our ablation studies systematically isolate and quantify the inductive biases contributed by each component of our hierarchical Bayesian nonparametric framework—namely, the nested variational family, structured variational inference, and stochastic quantization. Bayes-QLAE consistently outperforms all ablated variants across disentanglement metrics, demonstrating that each component provides complementary inductive biases that, when combined, enhance distinct aspects of disentanglement. Our empirical results demonstrate the effectiveness and generalizability of the proposed approach across image datasets characterized by diverse factor variation distributions. Bayes-QLAE consistently achieves superior or competitive performance relative to baseline methods on both 3DShapes and MPI3D, particularly in terms of both modularity and compactness-based disentanglement metrics. Importantly, this performance is attained solely through architectural inductive biases, without reliance on additional computationally expensive regularization.

The differential in performance on the two datasets suggests that the underlying distributional properties of generative factors significantly influence the efficacy of the nonparametric prior. We hypothesize that replacing the Dirichlet Process prior with a more flexible prior such as the Pitman–Yor process—which allows for a richer clustering structure and can model power-law behaviors—may yield further performance improvements. These findings highlight the potential of our framework for interpretable and structured representation learning in varied settings.

ACKNOWLEDGMENTS

This work was partially supported by the University of Maryland Institute for Health Computing (UM-IHC) and the University of Maryland Institute for Advanced Computer Studies (UMIACS). We thank Professor Yiannis Aloimonos for insightful discussions and helpful feedback.

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

# A APPENDIX

## A.1 PRELIMINARIES

For a theoretical foundation for our proposed approach, we first review the Dirichlet Process and its role as a nonparametric prior in mixture models, with particular emphasis on the natural clustering structure that emerges from this formulation. Subsequently, we discuss the quantization-based generative model, the VQ-VAE, which utilizes discrete latent representation spaces and review prior work leveraging this quantization-structure for learning disentangled representations.

### A.1.1 DIRICHLET PROCESS MIXTURE MODELS

We begin by considering nonparametric models, defined by an infinite-dimensional parameter space that fundamentally allows the model's complexity to adapt and grow with the data. These models are typically used as priors over distributions with broad support that encompasses the entire space of all possible distributions. The Dirichlet Process (DP), in particular, is a stochastic process whose realizations are discrete probability distributions, thereby defining a valid nonparametric prior probability distribution over the space of discrete probability measures. Sethuraman (1994) constructive definition of the DP represents each discrete distribution drawn from the DP as a weighted sum of countably infinite atomic measures sampled from a continuous base distribution. This definition uses the stick-breaking construction, where the infinite sequence of weights for the atomic measures of the discrete distribution is generated by iteratively partitioning a unit-length stick. In the first step, a segment of length $\beta_1$ is broken off the stick, where $\beta_1$ is drawn from a Beta distribution, $\beta_1 \sim \text{Beta}(1, \alpha)$, parameterized by $\alpha$, ensuring $\beta_1 \in (0, 1)$. This segment is assigned as the weight of the first atomic measure $\theta_1$, which is independently sampled from a base distribution $G_0(\lambda)$ with parameters $\lambda$. The remaining portion of the stick, with length $1 - \beta_1$, is then recursively partitioned in the same manner: at each step, a segment of length $\beta_i \sim \text{Beta}(1, \alpha)$, scaled to the length of the remaining stick given by $\prod_{j=1}^{i-1}(1-\beta_j)$, is broken off and assigned as the weight of an atomic measure $\theta_i$, drawn independently from the base distribution. This explicit stick-breaking construction generates a random discrete distribution $G \sim \text{DP}(\alpha, G_0)$ over the countably infinite atomic measures $\theta$ drawn from the base distribution $G_0$. Building on this, the DP serves as the nonparametric prior over the mixture components in the Dirichlet Process Mixture Models (DPMMs) Antoniak (1974). DPMMs generate data by first sampling a discrete distribution $G$ from the prior $\text{DP}(\alpha, G_0)$ using the stick-breaking construction and then using the set of atomic measures $\theta$ sampled from the base distribution $G_0$ to parameterize a data-generating distribution $F$. To generate each data point, we first sample a latent variable $z$ from the discrete distribution defined by the stick-breaking weights, then use the corresponding atomic measure $\theta_z$ to parameterize the data-generating distribution $F$, which is further used to draw the observation $x$:

$$\beta_k \mid \alpha \sim p(\beta_k \mid \alpha) = \text{Beta}(1, \alpha), \quad \theta_k \mid \lambda \sim p(\theta_k \mid \lambda) = G_0(\lambda), \quad \forall k \in \mathbb{N} \qquad (12)$$

$$z = k \mid \beta \sim p(z = k \mid \beta) = \beta_k \prod_{j=1}^{k-1}(1 - \beta_j), \quad x \mid \{z, \theta\} \sim p(x \mid z, \theta) = F(\theta_z) \qquad (13)$$

Because each realization $G$ drawn from the DP is a discrete distribution over the atomic measures $\theta$, the above data-generating process results in repeated parameter values for the data-generating function $F$. This effectively induces a partitioning of the data, where each partition or component corresponds to the data points generated with identical parameter values, allowing the generative

process to be interpreted as a mixture model. Consequently, this results in a hierarchical Bayesian framework, where the parameters of the data-generating distribution $F$ are sampled from a discrete probability distribution drawn from the DP. The joint distribution of the data $\{x_1, \ldots, x_N\}$ and the latent variables: stick-breaking lengths $\beta = \{\beta_1, \beta_2, \ldots\}$, component parameters $\theta = \{\theta_1, \theta_2, \ldots\}$ and assignment variables $\{z_1, \ldots, z_N\}$, factorizes hierarchically as follows:

$$p(x, z, \beta, \theta \mid \alpha, \lambda) = p(\beta \mid \alpha)\, p(\theta \mid \lambda)\, p(z \mid \beta)\, p(x \mid z, \theta) \tag{14}$$

The primary objective of the learning process is to infer the posterior distribution of the latent variables $\beta$, $\theta$ and $z$ conditioned on the observed data $x$ and the hyperparameters $\alpha$, $\lambda$, denoted by $p(\beta, \theta, z \mid x, \alpha, \lambda)$.

Computing the exact posterior over the latent variables given the observed data introduces dependencies among the variables. As a consequence evaluating the marginal likelihood of the data requires integrating over every possible latent configuration, making it intractable. In the nonparametric setting, such as under a Dirichlet Process (DP) prior, the posterior cannot be computed exactly and must be approximated. Wainwright et al. (2008) introduce a deterministic approach to approximate the intractable posterior with a simpler, tractable family of distributions by breaking certain dependencies among latent variables. They define a variational family $q_\nu$, parameterized by free parameters $\nu$, and optimize $\nu$ to minimize the Kullback–Leibler divergence between $q_\nu$ and the true posterior. Equivalently, this corresponds to maximizing the evidence lower bound (ELBO) on the log marginal likelihood of the data, as defined below:

$$\log p(x \mid \alpha, \lambda) \geq \mathbb{E}_{q_\nu}\left[\log p(x, \boldsymbol{e}, \boldsymbol{z}, \boldsymbol{\beta}, \boldsymbol{\theta} \mid \alpha, \lambda) - \log q_\nu(\boldsymbol{e}, \boldsymbol{z}, \boldsymbol{\beta}, \boldsymbol{\theta})\right] \tag{15}$$

### A.1.2 VECTOR QUANTIZATION FOR DISENTANGLEMENT

Next, we discuss the Vector Quantized-Variational AutoEncoder (VQ-VAE) Van Den Oord et al. (2017), a generative model which learns a discrete latent representation via vector quantisation (VQ). The VQ-VAE model discretizes the continuous encoder outputs $z_e(x)$ by mapping them to a discrete latent space consisting of a codebook with a finite set of $K$ embedding vectors $\{e_k\}_{k=1}^K$. The posterior distribution $q(z \mid x)$ of the latent variable $z$ is categorical over the embedding space, with probabilities determined by the Euclidean distances between the encoder output and the embedding vectors in the codebook. Samples drawn from this distribution index the set of embedding vectors, which are then passed as input to the decoder $z_q$ as follows:

$$q(z = k \mid x) = \begin{cases} 1 & \text{for } k = \arg\min_j \|z_e(x) - e_j\|_2, \\ 0 & \text{otherwise}, \end{cases} \tag{16}$$

$$z \sim q(z \mid x), \quad z_q(x) = e_z = e_k \tag{17}$$

To enable gradient propagation through the non-differentiable quantization step, a straight-through estimator is used, wherein gradients from the decoder are directly propagated back to the encoder output $z_e(x)$. The loss function used to train the VQ-VAE, defined in equation 18, consists of the reconstruction loss, jointly optimizing the encoder and decoder to maximize the evidence lower bound (ELBO) on the data log-likelihood. Assuming a uniform prior $p(z)$ and a deterministic posterior as in equation 16, the KL divergence of the ELBO simplifies to the constant $\log K$ and is ignored. The second term corresponds to the vector quantization loss, which updates the embedding vectors by moving them toward the encoder outputs. The third term is the commitment loss, encouraging the encoder outputs to remain close to the selected embeddings and thereby ensuring alignment between the encoder space and the embedding space.

$$L = -\log p(x \mid z_q(x)) + \|\mathrm{sg}[z_e(x)] - e\|_2^2 + \beta\|z_e(x) - \mathrm{sg}[e]\|_2^2, \tag{18}$$

where sg stands for the stop-gradient operator which blocks the gradient from propagating through the computational branch of the operand, treating it as a constant. While standard VQ-VAE approaches discretize the latent representations using a single codebook of high-dimensional embedding vector and optimize primarily for reconstruction fidelity, learning disentangled representations necessitates strong inductive biases Locatello et al. (2019b). To structure the latent space such that distinct dimensions capture independent generative factors, the approach of Hsu et al. (2024a) instead propose latent quantization, which enforces structural regularity in the latent space by quantizing each latent dimension using separate learnable scalar codebooks. Specifically, the proposed quantized latent autoencoder (QLAE) parameterizes the latent space as the Cartesian product

$Z = C_1 \times \cdots \times C_d$, where each codebook $C_j$ contains scalar embeddings. This element-wise quantization enforces a combinatorial factorized encoding, allowing the decoder to learn consistent interpretations for each latent dimension. Furthermore, a higher weight decay is used to regularize the model to encourage reliance on the discrete codebook structure. Collectively, these design choices promote disentangled representations through explicit architectural and regularization biases.

## A.2 Network Architectures and Hyperparameters

**Recognition Network (or Encoder)**.[1] The encoder is a feedforward convolutional network. Each convolutional block consists of a downsampling convolution (kernel size 4, stride 2) followed by batch normalization, and each block is followed by a Leaky ReLU activation (slope 0.3). Four such blocks with channel widths 32, 64, 128, and 256 produce a $256 \times 4 \times 4$ feature map. This feature map is flattened and passed through two fully connected layers of width 256 with Leaky ReLU activations. A final linear layer outputs the mean and precision parameters of the approximate conjugate likelihood, using a Softplus transformation on the precision to enforce positivity.

**Data-Generating Function (or Decoder)**. The sampled latent representation is first passed through two fully connected layers of width 256 with ReLU activations and no normalization, preserving the piecewise affine structure of the mapping. The resulting feature vector is reshaped to $256 \times 4 \times 4$ and processed by four transposed-convolution blocks. Each block applies an upsampling transposed convolution (kernel size 4, stride 2) followed by a Leaky ReLU activation (slope 0.3). The blocks use channel widths 128, 64, 32, and 3, yielding a final observation of shape $3 \times 64 \times 64$. For a binary cross entropy loss function the output is followed by a sigmoid activation function.

| hyperparameter | value |
|---|---|
| $d$ | 10 |
| $\alpha$ | 1 (with a Gamma(1, 1) prior) |
| $\beta$ (scaling of KL div.) | 0.005 |
| $\lambda = (mu_0, \kappa_0, \nu_0, w_0)$ | $(0., 1., 2., 0.1)$ |
| Autoencoder learning rate | $1 \times 10^{-4}$ |
| Latent learning rate | $1 \times 10^{-3}$ |
| Adam $\beta_1$ | 0.9 |
| Adam $\beta_2$ | 0.99 |
| Adam updates | $2 \times 10^5$ |
| Grad clip norm | 5.0 |
| batch size | 64 |

Table 3: Fixed Hyperparameters and initializations

## A.3 Ablation Studies and Discussion

| Model | Info M | Info C | Info E |
|---|---|---|---|
| Bayes-QLAE | $0.58 \pm .04$ | $0.51 \pm .03$ | $0.71 \pm .04$ |
| T-QLAE (k=10) | $0.54 \pm .04$ | $0.40 \pm .03$ | $0.68 \pm .04$ |
| T-QLAE (k=50) | $0.51 \pm .06$ | $0.48 \pm .05$ | $0.62 \pm .06$ |
| MF-QLAE | $0.49 \pm .04$ | $0.49 \pm .04$ | $0.76 \pm .04$ |
| DQ-QLAE | $0.52 \pm .02$ | $0.43 \pm .02$ | $0.71 \pm .02$ |

Table 4: Model performance comparison across different information metrics

We structure the experiments in this section to isolate and quantify the specific inductive biases derived from each component of our hierarchical Bayesian nonparametric framework: the nested variational family, structured variational inference, and stochastic quantization. Specifically, we

---

[1]The implementation is available https://github.com/vspatil-npbayes/bayes-qlae.git

perform an ablation by replacing the nested variational family with a truncated one (T-QLAE) with different truncation levels $K$. Similarly, to evaluate the role of the structured variational family, we substitute it with a mean-field variational family (MF-QLAE), as detailed in (Johnson et al., 2016). Finally, to isolate the effect of stochastic quantization, we replace with a deterministic nearest-neighbor quantization, with the straight-through estimator used to propagate gradients through the quantization step.

From our experiments (as detailed in Table 4) Bayes-QLAE consistently outperforms its ablated variants across all disentanglement metrics, confirming that each component contributes an inductive bias which, when combined, enhances performance. For models based on truncated variational families, we observe a negative correlation between modularity and truncation level, while a positive correlation with compactness. This aligns with the intuition that representations obtained with fewer quantized values are biased toward modularity. Notably, the truncated model with $k = 10$ surpasses QLAE in modularity due to the benefits of stochastic quantization and structured variational inference, though with a slight reduction in compactness as a consequence of stochasticity. Removing structured variational inference and defaulting to deterministic quantization degrades both modularity and compactness, with the mean-field family exhibiting a more severe decline in modularity, suggesting a bias towards representation which minimize the reconstruction cost over representations adhering to structured prior distribution. Finally, deterministic quantization exacerbates posterior collapse, leading to representations with a lower compactness metric.

We empirically demonstrate that a small codebook is not a prerequisite for disentanglement and therefore need not be fixed a priori. Critically, across all three axes of disentanglement assessment—informativeness (reconstruction fidelity), modularity (independence), and compactness (one-to-one factor-dimension correspondence)—we observe that disentanglement quality remains stable or improves as the codebook size expands adaptively in response to data complexity. These findings directly challenge the common assumption that small, fixed codebooks are necessary for learning disentangled representations.

Rather, the critical factors enabling disentanglement are two structural properties: (1) the implicit regularization effect induced by discrete latent encodings, and (2) the combinatorial composition of factor-specific codes to encode representations. During early training stages, when the codebook size is small, the encoder operates under a representational bottleneck that necessitates the construction of latent representations through combinatorial composition of the restricted set of available codes. This bottleneck implicitly regularizes the learning process, strongly biasing the encoder toward allocating disjoint, factor-specific codes to each factor-specific codebook. Consequently, the learned compositional structure mirrors the underlying generative process of the dataset, wherein the set of observations arise from the Cartesian product of discrete factor instantiations. This early-stage regularization effect establishes a foundation for disentanglement by enforcing a modular, compositional encoding scheme that respects the factorial structure of the data-generating distribution.

In our approach we initialize the nonparametric prior with a single code per codebook. The nested variational family provides a principled mechanism to increase the number of codes: new codes are instantiated if and only if their inclusion yields an improvement in the variational lower bound. This criterion ensures that capacity expansion occurs only when statistically justified by the data. Consequently, the model inherits the inductive bias of sparse codebooks while avoiding any explicit hard constraint on the upper bound of the cardinality of the codebooks. This adaptive regularization mechanism resolves the tension between early-stage structural learning and asymptotic expressiveness.

We validate this hypothesis through ablation studies comparing our nested variational inference framework against a truncated variant. In the truncated approach, we fix the number of mixture components at a predetermined upper bound for each factor, effectively eliminating the adaptive capacity of the nonparametric formulation. This modification results in measurable degradation across all disentanglement metrics relative to the nested variational inference approach. These results demonstrate that the adaptive, data-driven discovery of codebook size—rather than absolute codebook cardinality—is the essential mechanism underlying successful disentanglement.

Moreover, the tendency toward cluster expansion is explicitly governed by the concentration parameter $\alpha$ of the nonparametric prior, which is itself assigned a Gamma hyperprior. This hierarchical Bayesian formulation provides regularization of the cluster proliferation rate, enabling the model

to infer from data the appropriate balance between model parsimony and representational capacity without manual specification.

## A.4 DISENTANGLEMENT METRICS

For quantitative evaluation, we compute two complementary disentanglement metrics which comprehensively measure disentanglement properties using different computational approaches. The InfoMEC metric (Hsu et al., 2024a) relies on information-theoretic mutual information estimation computed from the empirical joint distribution between latent representations and ground-truth factors. In contrast, the DCI metric (Eastwood & Williams, 2018) trains predictive models to map the learned representations to the underlying factors of variation. Both metrics evaluate disentanglement quality across three fundamental dimensions, though with different terminology: InfoMEC measures InfoModularity (InfoM) while DCI measures Disentanglement (D) to quantify the extent to which sources are encoded in mutually disjoint subsets of representations, InfoExplicitness (InfoE) and Informativeness (I) measure the degree to which the relationship between the sources and representations can be characterized by a simple functional or statistical dependency, and InfoCompactness (InfoC) and Completeness (C) quantifies the degree to which latent variables encode information exclusively about mutually disjoint subsets of the sources. We train our proposed approach in a completely unsupervised manner on the entire dataset and evaluate the learned representations on a subset of samples, using the open-source implementations of disentanglement metrics by Locatello et al. (2019b); Hsu et al. (2024a).

## A.5 QUALITATIVE EVALUATION

We conduct qualitative assessments of our proposed method on each dataset to evaluate both sample reconstructions and latent traversals. For latent traversals, we encode a single image into the latent space and systematically visualize the effects of intervening on individual latent dimensions. Specifically, for each latent variable we vary its value across the range of values encoded in the representations (sample with replacement) while holding all other latent dimensions fixed, then decode the resulting latent vectors to observe their effects in the data space. In the visualization, each row corresponds to interventions on a single latent variable, while columns represent different values sampled from the empirical distribution of that dimension. Well-disentangled representations should exhibit smooth, semantically meaningful changes along individual latent dimensions, with each dimension controlling a distinct generative factor independently of others.

**Reconstruction Fidelity**. We assess the informativeness of learned representations by examining reconstruction quality. High-fidelity reconstructions that faithfully preserve visual details of the original images indicate that the latent representations are sufficiently informative to capture the full range of variations present in the data. Conversely, poor reconstructions suggest that certain factors of variation have been inadequately encoded or lost during the encoding process.

**Modularity**. We evaluate the modularity, or disentanglement, properties of learned representations through latent traversal analysis. A representation exhibits modularity when each latent dimension independently controls a single underlying generative factor while remaining invariant to variations in other factors. Operationally, this is assessed by examining whether each row in the traversal visualization demonstrates isolated semantic changes corresponding to a single factor of variation without coupling to other factors. Such independence in the latent space reflects successful recovery of the true compositional structure of the underlying independent generative factors.

**Compactness**. We further assess the compactness, or completeness, of the learned representations by determining whether all variations of a single generative factor are captured within a single latent dimension. Compact representations, wherein each factor is encoded by exactly one latent variable, are crucial for interpretability as they establish a one-to-one correspondence between latent dimensions and semantically meaningful generative factors. This property enables intuitive understanding and manipulation of specific attributes in the generated outputs.

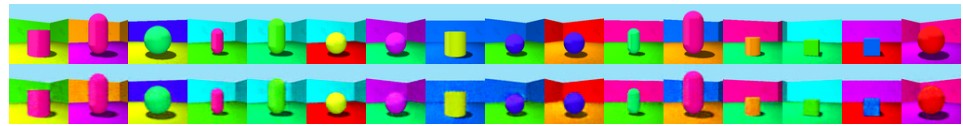

(a) Sample reconstructions: Original images (top row) and corresponding reconstructions (bottom row)

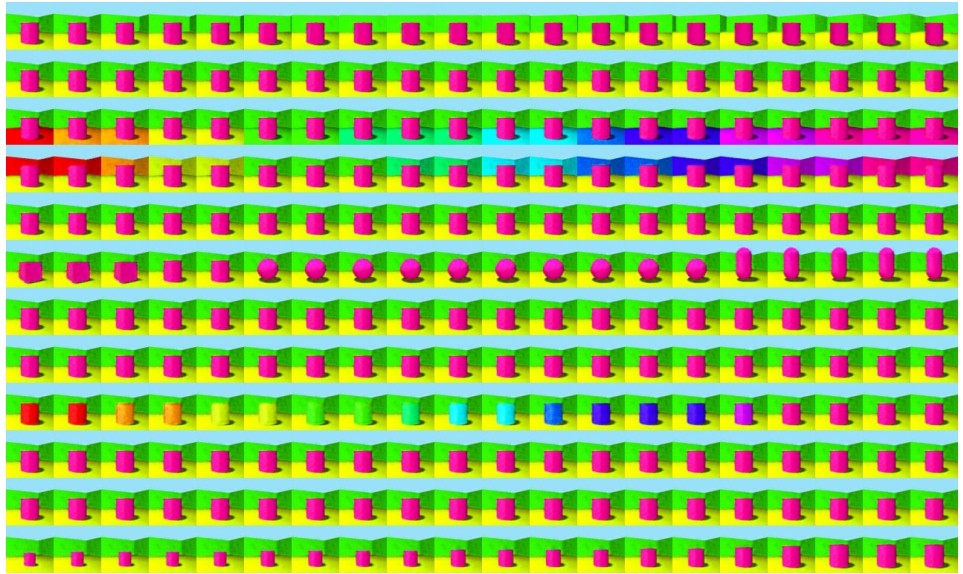

(b) Latent traversals: Each row shows the effect of systematically varying a single latent dimension while holding all other dimensions fixed. Columns represent different values sampled from the distribution of that dimension. The model successfully disentangles six ground-truth factors of variation: object orientation (row 1), floor hue (row 3), wall hue (row 4), object shape (row 6), object hue (row 9), and object scale (row 12). Rows 2, 5, 7, 8, 10, and 11 correspond to inactive latent dimensions that do not encode interpretable factors.

Figure 1: Reconstructions and Latent traversals for the 3DShapes dataset: Reconstructions demonstrate high fidelity in capturing visual details demonstrating the model's ability to faithfully encode and decode the full range of variations in the data. Latent traversals illustrate that individual latent variables control distinct, interpretable factors of variation in the generated images. Moreover, the presence of inactive latent variables and the encoding of each factor in a single latent variables indicates that the model has learned a compact representation recovering the true generative structure

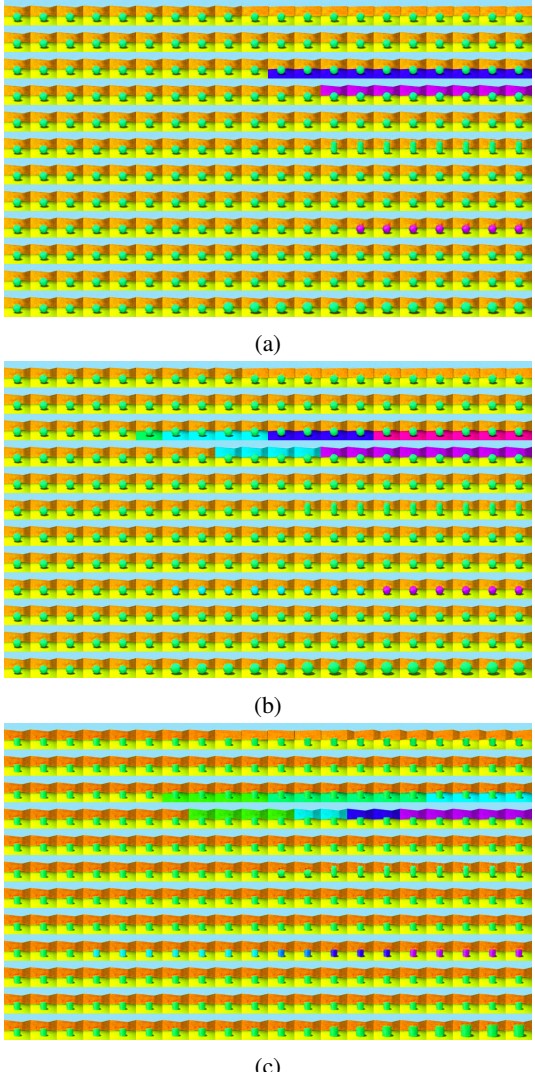

(a)

(b)

(c)

Figure 2: Evolution of Latent Traversals for 3DShapes dataset: The figure illustrates the evolution of the number of clusters associated with each generative factor demonstrating the adaptive capacity of the nonparametric formulation. Each column in each row corresponds to a factor-specific mixture component, and the distinct components within a row denote the clusters capturing the encoded variations of that factor. The vertical axis indicates cluster multiplicity, revealing how the model progressively discovers and encodes additional variations for each factor. This dynamic cluster growth exemplifies the nonparametric property of the hierarchical mixture prior, which enables data-driven inference of latent capacity without manual specification. Notably, factors with higher contribution to the reconstruction objective—such as floor hue, object hue, and wall hue—exhibit earlier cluster proliferation during training, suggesting the model prioritizes encoding variations that most significantly impact reconstruction fidelity. In contrast, geometric factors such as object orientation and shape undergo refinement in later training stages, indicating a hierarchical learning strategy wherein the model first captures high-variance attributes before refining lower-variance structural properties. This demonstrates that the nonparametric prior successfully balances model capacity across factors according to their respective complexities and contributions to data likelihood

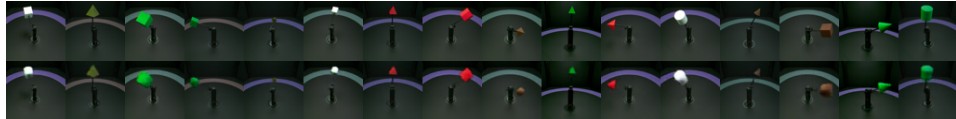

(a) Sample reconstructions: Original images (top row) and corresponding reconstructions (bottom row)

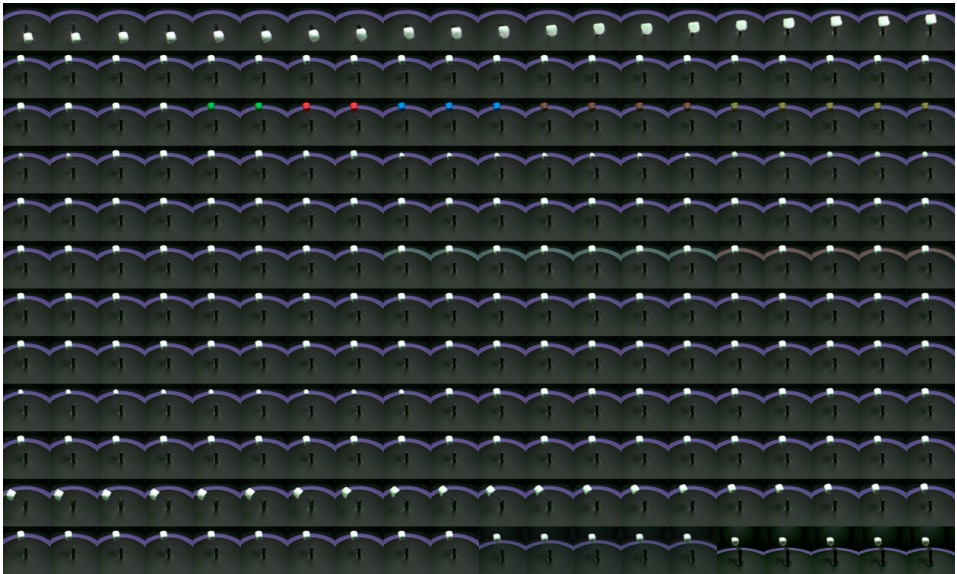

(b) Latent traversals: Each row shows the effect of systematically varying a single latent dimension while holding all other dimensions fixed. Columns represent different values sampled from the distribution of that dimension. The model successfully disentangles the following ground-truth factors of variation: vertical axis (row 1), object color (row 3), object shape (row 4), background color (row 6), object size (row 9), horizontal axis (row 11) and camera height (row 12). Rows 2, 5, 7, 8, 10 correspond to inactive latent dimensions that do not encode interpretable factors.

Figure 3: Reconstructions and Latent traversals for the MPI3D real dataset: Reconstructions demonstrate high fidelity in capturing visual details demonstrating the model's ability to faithfully encode and decode the an extensive range of variations in the data. Latent traversals illustrate that individual latent variables control distinct, interpretable factors of variation in the generated images. It is worth noting that, although the representations do not capture the full set of underlying variations, they remain both modular and compact, closely reflecting the true underlying structure of the generative process.

