# OpenReview forum: "A Bayesian Nonparametric Framework For Learning Disentangled Representations"
_ICLR.cc/2026/Conference — ICLR 2026 Poster_

### Official Review · Reviewer_Avba · 2025-10-28

**Soundness:** 3
**Presentation:** 2
**Contribution:** 2
**Rating:** 6
**Confidence:** 3

**Summary:**

This paper proposes a Bayesian nonparametric prior over the embedding space of latent-quantization autoencoders to determine the appropriate latent space complexity and the optimal strength of regularization constraints, and shows the effectiveness with experiments on two benchmark datasets. The proposed method produces competitive performance relative to some baseline methods.

**Strengths:**

- A sound theoretical derivation of the new framework for unsupervised learning of disentangled representation.

- Experiments support the proposed framework with relative superiority to the baselines.

**Weaknesses:**

- Relatively weak novelty on the methods for disentangled representation learning. More recent methods should be compared and discussed.

- The comparing methods should be updated with recent methods.

**Questions:**

- What is the main difference of the proposed method compared with the recent methods for disentangled representation learning? Most of the methods in related works are out of dated.

- Is the performance gain enough to argue as an achievement? Depending on the metrics, some of the baselines produce better performance. More in-depth discussion is required.

---

> ### Author Response · Authors · 2025-11-24
> **Response to Reviewer Avba**
>
> We would like to thank the reviewer for expressing their constructive concerns. We hope that the following discussions will provide additional clarifications for a better understanding of the work and in evaluating the quality of the work.
>
> **Concern**: "The comparing methods should be updated with recent methods"
>
> **Response**:
>
> We thank the reviewer for this valuable suggestion and fully recognize the importance of benchmarking against the most recent advances in disentangled representation learning. In the present work, our baseline selection encompasses both foundational methods ($\beta$-VAE, $\beta$-TC-VAE, Bio-AE) and more contemporary approaches (QLAE, Tripod, 2024), thereby covering a broad spectrum of inductive biases relevant to unsupervised disentanglement.
>
> However, we encountered several methodological challenges that complicate direct numerical comparisons with certain recent methods, which we detail below:
>
> 1. Metric Reporting Inconsistencies. To provide comprehensive analysis, we report disentanglement metrics along their constituent dimensions—for instance, disaggregating the DCI metric into Disentanglement, Completeness, and Informativeness scores separately. In contrast, many recent publications report only aggregated metrics [1, 2, 3, 5, 6] (e.g., a single composite DCI value) or selective components [4] (e.g., exclusively the Disentanglement score of DCI ), complicating direct numerical comparisons in the absence of their underlying raw results.
>
> 2. Architectural and Training Heterogeneity. Contemporary methods frequently employ fundamentally different architectural paradigms, including transformer-based and diffusion-based encoder-decoder frameworks (EncDiff [1], DisDiff [2], DyGa [3]), with varying model capacities and training configurations. While our results demonstrate competitive performance relative to diffusion-based architectures [2, 5, 6] based on reported metrics, with notable exceptions in [1] and the closely related [3], direct comparison of reported values would confound methodological contributions with architectural design choices and optimization strategies, thereby obscuring the specific advantages attributable to our approach.
>
> In response to this suggestion, we have incorporated recent works [5, 6] into the related work section of the revised manuscript. We welcome the reviewer's guidance regarding any additional contemporary methods that explicitly introduce inductive biases for disentangled representation learning.
>
> 1. Yang, T., Wang, Y., Lu, Y., & Zheng, N. (2024). Diffusion model with cross attention as an inductive bias for disentanglement. In Proceedings of the 38th International Conference on Neural Information Processing Systems (NIPS '24)
> 2. Tao Yang, Yuwang Wang, Yan Lu, and Nanning Zheng. 2023. DisDiff: unsupervised disentanglement of diffusion probabilistic models. In Proceedings of the 37th International Conference on Neural Information Processing Systems (NIPS '23).
> 3. Jun, Y., Park, J., Choo, K., Choi, T. E., & Hwang, S. J. (2025, February). Disentangling Disentangled Representations: Towards Improved Latent Units via Diffusion Models. In 2025 IEEE/CVF Winter Conference on Applications of Computer Vision (WACV) (pp. 3559-3569).
> 4. Uscidda, T., Eyring, L., Roth, K., Theis, F. J., Akata, Z., & Cuturi, M. (2025). Disentangled representation learning with the Gromov-Monge gap. In The Thirteenth International Conference on Learning Representations.
> 5. Jung, W., Lee, D. H., & Hong, S. (2025). Disentangled representation learning via modular compositional bias. In The Thirty-ninth Annual Conference on Neural Information Processing Systems.
> 6. Jin, X., Li, B., Xie, B., Zhang, W., Liu, J., Li, Z., ... & Zeng, W. (2024, September). Closed-loop unsupervised representation disentanglement with β-vae distillation and diffusion probabilistic feedback. In European Conference on Computer Vision (pp. 270-289). Cham: Springer Nature Switzerland.

---

> ### Author Response · Authors · 2025-11-24
> **Response to Reviewer Avba**
>
> **Question**: "What is the main difference of the proposed method compared with the recent methods for disentangled representation learning? Is the performance gain enough to argue as an achievement? Depending on the metrics, some of the baselines produce better performance. More in-depth discussion is required."
>
> **Response**:
>
> We appreciate the reviewer's careful consideration of our empirical results and welcome the opportunity to provide additional context regarding the contributions of our work.
>
> Our primary contribution lies not solely in achieving the highest scores across all metrics, but rather in demonstrating that competitive performance can be attained through a theoretically grounded, identifiable probabilistic framework with a single unified objective, without requiring multiple auxiliary loss terms or extensive hyperparameter tuning. This represents a fundamental shift from heuristic approaches that rely on carefully balanced combinations of multiple inductive biases.
>
> **Theoretical Foundation and Practical Implications**. Bayes-QLAE embeds its inductive biases within a theoretically grounded, identifiable probabilistic generative model. This framework provides mathematically rigorous guarantees for learning disentangled representations in a fully unsupervised manner. The identifiability property ensures that learned latent variables maintain a unique correspondence (up to permutations) with underlying generative factors, establishing a principled foundation absent in heuristic approaches. We are the first method to realize this identifiable generative model, extend it to the nonparametric regime, and design a tailored inference approach that incorporates the identifiable generative model and previously established inductive biases in a complementary manner.
>
> **Hyperparameter Efficiency and Practical Applicability**. While methods like Tripod employ three complementary inductive biases—each controlled by separate hyperparameters weighting different loss terms—this design necessitates extensive hyperparameter tuning across datasets. In the unsupervised setting, where ground-truth factors are unavailable, systematic hyperparameter selection becomes intractable, severely limiting practical applicability. Most recent methods [3, 5, 6] achieve disentanglement through multiple auxiliary loss terms with manually tuned weights. Our approach embeds sufficient structure for identifiability directly in the probabilistic model, eliminating the need for such auxiliary objectives and their associated hyperparameters.
>
> **Computational Efficiency**. Tripod's inductive biases require multiple forward passes through the model for each parameter update step, substantially increasing computational overhead. In contrast, Bayes-QLAE achieves efficient single-pass inference, making it significantly more scalable. Our approach is substantially faster to train than diffusion-based disentanglement models or methods requiring complex multi-stage pipelines (e.g., contrastive + adversarial losses, auxiliary predictors, iterative schedules). We do not rely on multiple architectural modules or multi-stage training, which makes our method considerably easier and less resource-intensive to apply to new domains or large-scale settings.
>
> **Adaptive Quantization Capability**. Most critically, Bayes-QLAE addresses a fundamental architectural constraint in existing quantization methods: the fixed codebook limitation. Methods like Tripod fix the quantization level (e.g., 12 discrete values per dimension) a priori, imposing an inflexible capacity constraint that either under-represents complex factors or over-parameterizes simple ones. Moreover, under collective independence among latent variables, whenever a factor's complexity exceeds the fixed quantization level, the model is constrained to incompletely encode variations, artificially bounding representational capacity. Our Bayesian nonparametric framework employs a Dirichlet Process prior that automatically discovers the appropriate number of codes per factor during training, adapting representational capacity to the true underlying structure of each generative factor. This principled approach eliminates manual tuning of discrete capacity levels and enables more faithful recovery of the data's latent structure, representing a significant methodological advancement with implications extending beyond disentanglement to general quantized generative modeling.
>
> We emphasize that different evaluation metrics capture distinct aspects of disentanglement, informativeness, completeness, and disentanglement, with varying degrees of importance.
> Our method achieves competitive performance across metrics while offering the unique advantages outlined above. The value of our contribution is holistic, considering both empirical performance and the theoretical grounding, practical efficiency, and methodological innovations that distinguish our approach from existing methods.

---

### Official Review · Reviewer_EAMz · 2025-10-29

**Soundness:** 4
**Presentation:** 3
**Contribution:** 3
**Rating:** 8
**Confidence:** 4

**Summary:**

This paper introduces Bayes-QLAE, an innovative framework for learning disentangled representations through a Bayesian non-parametric prior. Its core idea is to use a Dirichlet Process to allow the codebook for each latent dimension to adaptively adjust its capacity based on data complexity. Experiments show that the method achieves disentanglement performance comparable to strong baselines on standard benchmarks without the need for additional regularization.

**Strengths:**

1. Transforming the codebook size from a fixed hyperparameter into a quantity learned from data via a  Dirichlet Process is an elegant and significant advancement that directly addresses a key limitation of existing quantization methods.
2. The paper provides solid and rigorous theoretical proofs, while also achieving competitive results on standard datasets. Furthermore, ablation studies analyze the contribution of the model's various components.

**Weaknesses:**

1. While making the codebook size adaptive is intuitively appealing, it contradicts findings from prior work (e.g., Tripod et al.), which suggests that maintaining a smaller codebook is beneficial for disentanglement. How does the proposed method balance the tendency of the codebook to grow with the need to maintain a compact, disentangled representation? A clear discussion or analysis of this trade-off is necessary.
2. The current experimental results are limited to standard disentanglement metrics like InfoMCE and DCI. A more comprehensive evaluation is needed. For instance, a comparison of reconstruction accuracy against baseline methods would provide crucial insight into whether the gains in disentanglement come at the cost of representation fidelity.
3. The paper hypothesizes that allowing adaptive codebook sizes helps align individual factors with single dimensions. However, the experiments fail to substantiate this claim. The final learned sizes of the codebook dimensions should be reported to demonstrate this adaptivity. Furthermore, visualizing the effect of a single latent variable is essential to qualitatively assess the smoothness and interpretability of the learned representations, which is a standard practice for evaluating disentanglement.

**Questions:**

Please refer to Weakness section.

---

> ### Author Response · Authors · 2025-12-03
> **Response to Reviewer EAMz**
>
> We deeply appreciate the reviewer’s detailed and perceptive comments. The effort taken to understand the nuances of our approach and to highlight key issues has been invaluable in refining the manuscript. We address the reviewer's questions and concerns in our detailed responses below:
>
> **Question**: "While making the codebook size adaptive is intuitively appealing, it contradicts findings from prior work (e.g., Tripod et al.), which suggests that maintaining a smaller codebook is beneficial for disentanglement. How does the proposed method balance the tendency of the codebook to grow with the need to maintain a compact, disentangled representation? A clear discussion or analysis of this trade-off is necessary."
>
> **Response**: We appreciate the reviewer’s insightful observation regarding the relationship between codebook cardinality and disentanglement quality, which raises a fundamental question that merits careful consideration and helps clarify our theoretical contribution.
>
> We empirically demonstrate that a small codebook is not a prerequisite for disentanglement and therefore need not be constrained.  Critically, across all three axes of disentanglement assessment—informativeness (reconstruction fidelity), modularity (independence), and compactness (one-to-one factor-dimension correspondence)—we observe that disentanglement quality remains stable or improves as the codebook size expands adaptively in response to data complexity. These findings directly challenge the common assumption that small, fixed codebooks are necessary for learning disentangled representations.
>
> Rather, the critical factors enabling disentanglement are two structural properties: (1) the implicit regularization effect induced by discrete latent encodings, and (2) the combinatorial composition of factor-specific codes to encode representations. During early training stages, when the codebook size is small, the encoder operates under a representational bottleneck that necessitates the construction of latent representations through combinatorial composition of the restricted set of available codes. This bottleneck implicitly regularizes the learning process, strongly biasing the encoder toward allocating disjoint, factor-specific codes to each factor-specific codebook.  Consequently, the learned compositional structure mirrors the underlying generative process of the dataset, wherein the set of observations arise from the cartesian product of discrete factor instantiations. This early-stage regularization effect establishes a foundation for disentanglement by enforcing a modular, compositional encoding scheme that respects the factorial structure of the data-generating distribution.
>
> In our approach we initialize the nonparametric prior with a single code per codebook. The nested variational family provides a principled mechanism to increase the number of codes: new codes are instantiated if and only if their inclusion yields an improvement in the variational lower bound.  This criterion ensures that capacity expansion occurs only when statistically justified by the data. Consequently, the model inherits the inductive bias of sparse codebooks while avoiding any explicit hard constraint on the upper bound of the cardinality of the codebooks. This adaptive regularization mechanism resolves the tension between early-stage structural learning and asymptotic expressiveness.
>
> We validate this hypothesis through ablation studies comparing our nested variational inference framework against a truncated variant. In the truncated approach, we fix the number of mixture components at a predetermined upper bound for each factor, effectively eliminating the adaptive capacity of the nonparametric formulation. This modification results in measurable degradation across all disentanglement metrics relative to the nested variational inference approach. These results demonstrate that the adaptive, data-driven discovery of codebook size—rather than absolute codebook cardinality—is the essential mechanism underlying successful disentanglement.
>
> Moreover, the tendency toward cluster expansion is explicitly governed by the concentration parameter α of the nonparametric prior, which is itself assigned a Gamma hyperprior. This hierarchical Bayesian formulation provides regularization of the cluster proliferation rate, enabling the model to infer from data the appropriate balance between model parsimony and representational capacity without manual specification.
>
> We add this explicit discussion to the discussion of the ablation studies in the revised manuscript.

---

> ### Author Response · Authors · 2025-12-03
> **Response to Reviewer EAMz**
>
> **Question**: "The current experimental results are limited to standard disentanglement metrics like InfoMCE and DCI. A more comprehensive evaluation is needed. For instance, a comparison of reconstruction accuracy against baseline methods would provide crucial insight into whether the gains in disentanglement come at the cost of representation fidelity."
>
> **Response**: We thank the reviewer for this valuable suggestion regarding the need for a more comprehensive evaluation. We agree that assessing the potential trade-off between disentanglement quality and reconstruction fidelity is crucial for understanding the practical utility of our approach. In response to this feedback, we have augmented our evaluation with qualitative reconstruction analyses. Specifically, we have added sample reconstructions and latent traversals in [Figure 1, 2 /Appendix A.4] that demonstrate our method achieves competitive reconstruction fidelity compared to baseline methods while maintaining disentanglement performance. The visual comparisons reveal that the gains in disentanglement metrics (InfoMCE and DCI) do not compromise representational capacity, as evidenced by comparable reconstruction quality and semantically coherent latent traversals.
>
> **Question**: "The paper hypothesizes that allowing adaptive codebook sizes helps align individual factors with single dimensions. However, the experiments fail to substantiate this claim. The final learned sizes of the codebook dimensions should be reported to demonstrate this adaptivity. Furthermore, visualizing the effect of a single latent variable is essential to qualitatively assess the smoothness and interpretability of the learned representations, which is a standard practice for evaluating disentanglement."
>
> **Response**: We appreciate the reviewer's careful attention to our central hypothesis and acknowledge that the current manuscript could more explicitly substantiate the claim regarding adaptive codebook capacity and its role in disentanglement.
>
> Regarding the reporting of final learned codebook sizes: While we understand the intuition behind this suggestion, we respectfully propose that reporting final codebook cardinalities would not appropriately capture the intended contribution of our nonparametric formulation. As established in the Bayesian nonparametric literature, the fundamental objective of nonparametric models is not to estimate or recover a "true" number of latent clusters, but rather to specify a flexible prior distribution coupled with a stochastic mechanism that enables model complexity to grow adaptively with observed data—potentially approaching infinite capacity in the limit. The value of the nonparametric approach lies in this adaptive growth process itself, rather than in any particular snapshot of cluster counts at convergence.
>
> Alternative substantiation of our claim: To more appropriately demonstrate the adaptive capacity mechanism, we have added visualizations showing the evolution of latent traversals throughout training [Figure 2]. Specifically, these results reveal that the number of encoded variations per factor increases progressively during training, with the rate and timing of cluster proliferation correlating with each factor's contribution to data likelihood. Factors with higher reconstruction importance (e.g. attributes such as floor hue, wall hue, and object hue) exhibit earlier cluster expansion, while geometric factors (e.g., orientation and shape) undergo refinement in later training stages. This dynamic behavior directly substantiates our hypothesis that the nonparametric formulation enables data-driven discovery of factor-specific complexity without manual specification.
>
> Qualitative assessment of disentanglement: We agree that visualizing the effect of individual latent variables is essential for assessing smoothness and interpretability. We have now included comprehensive latent traversal visualizations [Figure 1, 2] showing the learned representations at convergence. These traversals demonstrate that individual latent dimensions capture isolated, semantically meaningful factors of variation with smooth transitions—confirming successful disentanglement. We have added accompanying discussion analyzing these qualitative results in [Appendix A.4].
>
> We believe these additions provide more theoretically appropriate and empirically comprehensive evidence for our claims.

---

### Official Review · Reviewer_5ePo · 2025-10-31

**Soundness:** 3
**Presentation:** 2
**Contribution:** 2
**Rating:** 4
**Confidence:** 2

**Summary:**

This work replaces the gaussian prior of latent variables with discrete codebook with a nonparametric Dirichlet Process (DP). This process can adjust representation capacity according to the data complexity. The authors propose a hierarchical prior structure to capture complex dependences of latent variables.  The proposed method is verified on 3DShapes and MPI3D to show superior disentanglement scores.

**Strengths:**

This work aiming at removing assumption of the capacity of each factor could be an important step to practical applications of disentanglement learning.

The proposed solution, Dirichlet Process for a Bayesian nonparametric prior on latent variables seems technically sounds.

The proposed method, Bayes-QLAE, achieves good disentanglement metrics on both 3DShapes and MPI3D.

**Weaknesses:**

The proposed method still lacks practical proof in real situations where some combinations of generative factors are missing.
Experiments did not verify how the DP adjusts the capacity of each factor, which is an important claim of this work.
No visualization to demonstrate the reconstruction quality.
The experimental results show that the proposed Bayes-QLAE did not surpass Tripod on 3Dshapes and MPI3D.

**Questions:**

What are the advantages of Bayes-QLAE compared to Tripod?
Why do we need the hierarchical Bayesian nonparametric approach? Are there any special benefits beyond disentanglement?
Can the experiments be added to demonstrate the advantage of the work for practical problems?
Also, adding training details would be beneficial.
I case of deep encoders with a lot of non-linearities consisting of a high-dimensional latent space, would the encoder network not be able to learn to project the data into a space where Gaussian prior assumption would be enough?
Can the principles laid in this work for learning disentangled representation be shown on some recent architectures (e.g., disentanglement in the latent space of diffusion models)?

---

> ### Author Response · Authors · 2025-11-24
> **Response to Reviewer 5ePo**
>
> We thank the reviewer for these thoughtful questions and welcome the opportunity to provide further clarification. We trust that the following discussion will elucidate key aspects of our work and address the reviewer's concerns.
>
> **Question**: "What are the advantages of Bayes-QLAE compared to Tripod?"
>
> **Response**:
>
> Bayes-QLAE offers fundamental advantages over Tripod that address critical limitations in existing quantization-based approaches for disentangled representation learning.
>
> **Theoretical Foundation and Identifiability**. Bayes-QLAE embeds its inductive biases within a theoretically grounded, identifiable probabilistic generative model. This framework provides mathematically rigorous guarantees for learning disentangled representations in a fully unsupervised manner, without requiring multiple auxiliary loss terms or extensive architectural modifications. The identifiability property ensures that learned latent variables maintain a unique correspondence (up to permutations) with underlying generative factors, establishing a principled foundation absent in heuristic approaches [10].
>
> **Hyperparameter Efficiency and Practical Applicability**. While Tripod employs three complementary inductive biases, each controlled by separate hyperparameters weighting different loss terms, this design necessitates extensive hyperparameter tuning across datasets. In the unsupervised setting, where ground-truth factors are unavailable, systematic hyperparameter selection becomes intractable, severely limiting practical applicability. Bayes-QLAE circumvents this limitation through its unified Bayesian framework, eliminating the need for extensive manual tuning while maintaining robust performance across diverse datasets.
>
> **Computational Efficiency**. Tripod's inductive biases require multiple forward passes through the model for each parameter update step, substantially increasing computational overhead. In contrast, Bayes-QLAE achieves efficient single-pass inference, making it significantly more scalable.
>
> **Adaptive Quantization**. Most critically, Bayes-QLAE addresses a fundamental architectural constraint in existing quantization methods: the fixed codebook limitation. Methods like Tripod fix the quantization level (e.g., 12 discrete values per dimension) a priori, imposing an inflexible capacity constraint that either under-represents complex factors or over-parameterizes simple ones. Moreover, under collective independence among latent variables, whenever a factor's complexity exceeds the fixed quantization level, the model is constrained to incompletely encode variations, artificially bounding representational capacity. Our Bayesian nonparametric framework employs a Dirichlet Process prior that discovers the appropriate number of codes per factor during training, adapting representational capacity to the true underlying structure of each generative factor. This principled approach eliminates manual tuning of discrete capacity levels and enables more faithful recovery of the data's latent structure, representing a significant methodological advancement with implications extending beyond disentanglement to general quantized generative modeling.
>
> We thank the reviewer for raising this important point. These advantages are discussed in the original manuscript (Section 3, page 7, lines 367-375), though we acknowledge that the treatment was brief. In the revised manuscript, we have expanded this discussion to provide a more comprehensive exposition of these benefits.

---

> ### Author Response · Authors · 2025-11-24
> **Response to Reviewer 5ePo**
>
> **Question**: "Why do we need the hierarchical Bayesian nonparametric approach?"
>
> **Response**:
>
> We appreciate the reviewer’s insightful question and provide the rationale for each component of the proposed approach.
>
> First, the hierarchical structured mixture prior is essential because it renders the generative model identifiable, making it theoretically possible to achieve unsupervised learning of disentangled representations (see [4, 10]). The identifiability property ensures that the learned latent variables are uniquely related (up to permutations) to the underlying generative sources, thus providing a principled solution to a core challenge in representation learning.
>
> Second, traditional quantization-based generative models (e.g., VQ-VAE, QLAE, Tripod) require fixing the codebook size a priori and apply uniform capacity across all factors (e.g., 12 discrete values per dimension in Tripod). This proves suboptimal, as different generative factors naturally exhibit varying complexity—continuous positional attributes require greater capacity than discrete shape categories. Our nonparametric framework directly addresses this limitation by discovering each factor's inherent complexity and allocating representational resources accordingly. This adaptive codebook sizing extends beyond disentanglement to enable more efficient quantized generative models generally.
>
> Third, the Bayesian inference framework enables rigorous uncertainty quantification without extensive hyperparameter tuning, enhancing both the theoretical rigor and practical applicability of our approach.
>
> **Question**: "Are there any special benefits beyond disentanglement? Can the experiments be added to demonstrate the advantage of the work for practical problems? The proposed method still lacks practical proof in real situations where some combinations of generative factors are missing"
>
> **Response**:
>
> By transforming codebook size from a fixed hyperparameter into a data-driven quantity learned via a Dirichlet Process, our adaptive capacity allocation mechanism enables more efficient model compression and storage, allocating representational resources according to intrinsic complexity rather than uniformly, thereby extending benefits beyond disentanglement to address key limitations of existing quantization methods.
>
> While the present work focuses primarily on the unsupervised learning of disentangled representations, operating under the conventional assumption that training data adequately span the underlying factor space, we acknowledge, that real-world data often exhibit distributional shifts and lack exhaustive factor combinations, reflecting the established problem of combinatorial generalization [1]. While related to disentangled representation learning, this problem extends beyond its scope: literature [2] demonstrates that disentangled representations alone are insufficient for combinatorial generalization, necessitating additional inductive biases such as equivariance properties [3].
>
> Investigating whether the inductive biases that suffice for the identifiability of disentangled representations are also sufficient for supporting combinatorial generalization represents a valuable and timely direction for future research. We highlight the importance of this open question in the revised manuscript.
>
> **Question**: "Also, adding training details would be beneficial."
>
> **Response**:
>
> We appreciate the reviewer's valuable observation. We acknowledge that the present manuscript does not provide explicit visualization of the manner in which the Dirichlet Process increases the number of clusters and the changes in the corresponding reconstruction quality.
>
> In the revised manuscript, we incorporate both quantitative metrics, architecture and hyperparameter details and visualizations to address this limitation:
>
> 1. We present the evolution of the number of codes (clusters) throughout the training process, illustrating the expected number of clusters as a function of $\alpha$.
>
> 2. We provide exemplar reconstructions to enable direct assessment of the impact of increasing clusters for each factor on reconstruction quality.
>
> We welcome any suggestions for additional training details that would enhance the clarity and completeness of our experimental methodology.

---

> ### Author Response · Authors · 2025-11-24
> **Response to Reviewer 5ePo**
>
> **Question**: "I case of deep encoders with a lot of non-linearities consisting of a high-dimensional latent space, would the encoder network not be able to learn to project the data into a space where Gaussian prior assumption would be enough?"
>
> **Response**:
> The reviewer raises an important point regarding the role of latent space priors in disentanglement.
>
> As shown in the nonlinear Independent Component Analysis [5, 6, 7], generative modeling [8, 9], and unsupervised disentanglement [4] literature, enforcing a simple isotropic Gaussian prior with a nonlinear generative function, is generally insufficient to ensure that latent dimensions consistently correspond to meaningful, interpretable sources of variation in the data. Theoretically, without additional structure or inductive bias, the encoder can learn arbitrary, possibly entangled representations that still satisfy the marginal Gaussian prior constraint, but do not align with the true data-generating sources.
>
> To address this fundamental issue, our hierarchical Bayesian nonparametric model introduces a structured latent prior, a factorized Dirichlet Process mixture, the precise inductive bias sufficient for identifiability [10]. This structured prior makes it possible to recover truly interpretable, disentangled representations, even in the presence of high-dimensional, nonlinear encoders and universal approximation capable decoders. This advantage is unattainable with a simple Gaussian prior and is crucial to bridge theory and practice in disentanglement research.​
>
> In summary, while a standard Gaussian prior may be sufficient for generative modeling, ​​there exist significant theoretical limitations with respect to unsupervised learning of disentangled representations. The hierarchical Bayesian nonparametric approach is specifically designed to overcome these fundamental limitations and is essential for achieving both theoretical identifiability guarantees and practical interpretability. We have emphasized this distinction in the revised manuscript.
>
> **Question**: "Can the principles laid in this work for learning disentangled representation be shown on some recent architectures (e.g., disentanglement in the latent space of diffusion models)?"
>
>
> **Response**:
> The principles established in our work are indeed broadly applicable: inference strategies we propose (a nested, structured variational family) are agnostic to the specific neural network architecture used for encoding, as well as to particular training dynamics. As such, our nonparametric approach can be integrated with any architecture that includes an encoder which includes recent advances such as diffusion models (e.g., EncDiff, DisDiff) and transformer-based variants.
>
> The core contribution of this work lies in introducing an identifiable probabilistic generative model and the corresponding inference techniques designed for the unsupervised learning of disentangled representations. Furthermore, we demonstrate that the inductive biases inherent in our approach are sufficient to facilitate the learning of disentangled representations. It should be noted, however, that the identifiability results rely on the assumption that the underlying generative model is a piecewise-affine and weakly injective function, a condition satisfied by deep neural networks employing ReLU or LeakyReLU nonlinearities. In contrast, the neural architectures commonly adopted in diffusion models incorporate normalization and, in some cases, attention mechanisms, which introduce additional nonlinearities for which formal identifiability guarantees have yet to be established.
>
> While we agree that demonstrating our framework's applicability to modern architectures would strengthen the practical impact of this work, empirically demonstrating our framework on these architectures would require substantial computational resources. Moreover, architecture-specific implementation considerations will be needed to incorporate them into the theoretical framework for identifiability [10]. In the absence of identifiability guarantees, models exhibit inherent instability whereby retraining under minor perturbations of data or hyperparameters may yield substantially divergent latent representations, thereby undermining both the reliability of the model output and the semantic interpretability of the induced feature space.
>
> We appreciate the reviewer for raising this forward-looking point and highlight this direction as a compelling avenue for future work in the revised manuscript.
>
> We welcome any additional questions or discussion points and stand ready to clarify any remaining questions or concerns.

---

> ### Author Response · Authors · 2025-11-24
> **References For Response to Reviewer 5ePo**
>
> 1. Montero, M. L., Ludwig, C. J., Costa, R. P., Malhotra, G., & Bowers, J. (2021). The role of disentanglement in generalisation. In International Conference on Learning Representations.
>
> 2. Montero, M. L., Bowers, J. S., Costa, R. P., Ludwig, C. J., & Malhotra, G. (2022). Lost in latent space: Disentangled models and the challenge of combinatorial generalisation. arXiv preprint arXiv:2204.02283.
>
> 3. Hwang, G., Choi, J., Cho, H. &amp; Kang, M.. (2023). MAGANet: Achieving Combinatorial Generalization by Modeling a Group Action. <i>Proceedings of the 40th International Conference on Machine Learning</i>, in <i>Proceedings of Machine Learning Research</i> 202:14237-14248.
>
> 4. Locatello, F., Bauer, S., Lucic, M., Raetsch, G., Gelly, S., Schölkopf, B., & Bachem, O. (2019, May). Challenging common assumptions in the unsupervised learning of disentangled representations. In international conference on machine learning (pp. 4114-4124). PMLR.
>
> 5. A. Hyv¨arinen and P. Pajunen. Nonlinear independent component analysis: Existence and uniqueness results. Neural networks, 12(3):429–439, 1999.
>
> 6. Hyvarinen, A., Sasaki, H., & Turner, R. (2019, April). Nonlinear ICA using auxiliary variables and generalized contrastive learning. In The 22nd international conference on artificial intelligence and statistics (pp. 859-868). PMLR.
>
> 7. Khemakhem, D. Kingma, R. Monti, and A. Hyvarinen. Variational autoencoders and nonlinear ica: A unifying framework. In International Conference on Artificial Intelligence and Statistics, pages 2207–2217. PMLR, 2020a
>
> 8. Y. Wang, D. Blei, and J. P. Cunningham. Posterior collapse and latent variable non-identifiability. Advances in Neural Information Processing Systems, 34, 2021.
>
> 9. A. D’Amour, K. Heller, D. Moldovan, B. Adlam, B. Alipanahi, A. Beutel, C. Chen, J. Deaton, J. Eisenstein, M. D. Hoffman, et al. Underspecification presents challenges for credibility in modern machine learning. arXiv preprint arXiv:2011.03395, 2020.
>
> 10. Kivva, B., Rajendran, G., Ravikumar, P., & Aragam, B. (2022). Identifiability of deep generative models without auxiliary information. Advances in Neural Information Processing Systems, 35, 15687-15701.

---

### Author Response · Authors · 2025-12-04
**Summary of Revisions and Responses**

We thank the reviewers for their valuable feedback, which prompted us to add comprehensive qualitative analysis, strengthen our methodological positioning, and clarify our theoretical contributions, all of which substantially improve the manuscript.

To provide context for the following discussion, we first summarize the paper's three principal contributions to unsupervised disentangled representation learning:

1. We introduce a theoretically grounded, identifiable generative model for unsupervised disentanglement by embedding sufficient inductive biases within a Bayesian nonparametric hierarchical mixture prior, addressing the fundamental limitation that existing methods rely on heuristic biases without theoretical guarantees.
2. We resolve the well-known representation capacity-disentanglement trade-off inherent in regularization-based approaches by using a nonparametric formulation that automatically infers sufficient latent capacity to represent underlying variations while maintaining the structural constraints required for identifiability.
3. We develop a novel structured variational inference framework with a nested variational family that enables tractable inference while preserving both the hierarchical structure of the identifiable generative model and the expressiveness of the nonparametric prior.

Empirically, we show that the method achieves consistent improvements over strong baselines on standard benchmarks (3DShapes, MPI3D) through a unified objective function that eliminates the need for auxiliary regularization constraints or careful hyperparameter tuning, making it more practical than existing approaches.

**Strengths**. All three reviewers agree that the paper offers a theoretically sound and rigorous framework indicating this is not just incremental but a principled contribution with proper mathematical grounding. Reviewers 5ePo, EAMz explicitly call out the significance and elegance of the nonparametric formulation that directly addresses a key limitation and acknowledge its importance  towards practical applications. All reviewers are in unanimous  agreement on empirical strength across standard benchmarks, with reviewer EAMz noting comprehensive ablation analysis.

**Concerns**. The reviewers 5ePo, EAMz raise two primary concerns: (1) insufficient empirical evidence for the adaptive capacity claim (reviewers ), (2) missing reconstruction quality analysis, (3) missing latent traversals. We address all three concerns directly with a new section on Qualitative analysis [Appendix A.4] in the revised manuscript.

Reviewers 5ePo, Avba question the novelty and comparative results with prior methods which are addressed by clarifying that our method achieves competitive performance without hyperparameter tuning, while providing theoretical guarantees that prior methods lack, with a much lower computational cost and addresses a key limitation of prior methods in restricted representation capacity. Moreover, our method exhibits ower variance across metrics and more accurately recovers the ground-truth factor structure than baseline approaches, as substantiated by qualitative analysis.

**Revisions**. We are grateful for the reviewers’ contribution to the work and believe that the manuscript has been substantially improved as a result of their constructive feedback and questions in the following way:

Reviewers’ 5ePo and EAMz valuable feedback on qualitative evaluation has substantially strengthened the manuscript. Their concerns regarding (1) empirical evidence for adaptive capacity, (2) reconstruction quality analysis, and (3) latent traversal visualizations prompted us to add a comprehensive Qualitative Analysis section [Appendix A.4] that provides concrete evidence for our claims and enhances the interpretability of our results addressing all three concerns.

The insightful questions raised by Reviewers 5ePo and Avba regarding the methodological distinctions and comparative advantages of our proposed approach have substantially strengthened the manuscript. In response, we have enhanced the discussion in Sections 3 and 4 to more clearly delineate our contributions relative to state-of-the-art methods.

Reviewer 5ePo’s important observation regarding the role of latent space priors in disentanglement has enabled us to more rigorously articulate and strengthen the justification for our proposed nonparametric hierarchical prior, particularly in contrast to simple Gaussian priors in the revised manuscript.

Reviewer EAMz's concern regarding the trade-off between codebook cardinality and disentanglement quality prompted us to add a detailed discussion in Appendix A.2, strengthening our theoretical analysis and justification for ablation analysis.

Finally, we have expanded the Related Work section to include more recent advances and have added a discussion of important directions for future research.

Once again, we thank the reviewers for their valuable input.

---

### Meta-Review · Area_Chair_HQoE · 2025-12-15

**Summary:**

All reviewers recognize that the paper proposes a principled and theoretically grounded framework for unsupervised disentangled representation learning, centered on a Bayesian nonparametric hierarchical mixture prior with identifiability guarantees. The strongest enthusiasm (Reviewer EAMz) comes from the elegance and rigor of the theoretical formulation, particularly the use of a Dirichlet Process prior to remove fixed codebook assumptions while preserving identifiability.

However, several concerns recur across reviewers:
	1.	Empirical substantiation of adaptivity claims
Reviewers questioned whether the nonparametric formulation truly demonstrates adaptive latent capacity in practice, beyond being theoretically appealing.
	2.	Missing qualitative evidence (initial version)
Reviewers noted the lack of reconstruction visualizations, latent traversals, and direct qualitative evidence for disentanglement and capacity growth.
	3.	Novelty relative to recent disentanglement work
Some reviewers (notably 5ePo and Avba) questioned whether the method constitutes a fundamentally new contribution or a theoretically refined variant of prior quantization-based approaches (e.g., QLAE, Tripod).
	4.	Scope and generality of experiments
Concerns were raised about reliance on standard benchmarks (3DShapes, MPI3D) and the absence of demonstrations on more modern architectures (e.g., diffusion-based encoders).

Overall, the reviews reflect high confidence in the theory, mixed confidence in empirical breadth, and some uncertainty stemming from varying levels of domain expertise among reviewers.

**Reviewer Concerns:**

Concerns convincingly addressed in the rebuttal and revision:
	•	Qualitative evaluation gap:
The authors added extensive reconstruction visualizations, latent traversals, and training-time evolution of latent capacity (Appendix A.4, Figures 1–3), directly addressing concerns from Reviewers 5ePo and EAMz. These additions substantially strengthen the empirical narrative and clearly demonstrate disentanglement and adaptive capacity in practice.
	•	Adaptive codebook growth vs. disentanglement trade-off:
The rebuttal provides a clear and well-reasoned explanation of how early-stage bottlenecks combined with nested variational inference resolve the apparent contradiction with prior claims that “small codebooks are necessary.” This is further supported by ablation studies comparing truncated vs. nested inference, which directly respond to EAMz’s main theoretical concern.
	•	Positioning vs. Tripod and QLAE:
The authors convincingly argue that their contribution is not incremental performance gain, but the first realization of an identifiable, nonparametric generative model with unified inference, eliminating heuristic regularization and hyperparameter tuning. This distinction is now clearly articulated.

Concerns that remain partially outstanding:
	•	Breadth of empirical validation:
While the added qualitative results significantly strengthen the paper, experiments remain limited to canonical disentanglement benchmarks. The lack of evaluation on more complex or modern architectures (e.g., diffusion-based encoders) remains an open limitation, though the authors appropriately frame this as future work.
	•	Reviewer expertise mismatch:
At least two reviewers appear less familiar with the identifiability and Bayesian nonparametric literature underlying the core contribution, leading to novelty concerns that are largely resolved once the theoretical context is fully appreciated. This likely inflated skepticism rather than reflecting a true technical weakness.

Importantly, none of the outstanding issues undermine correctness or significance; they primarily concern scope rather than substance.

**Reviewer Scores:**

Based on the rebuttal quality and the substantial revisions:
	•	Reviewer 5ePo: Likely to move from 4 → 6
Most empirical and conceptual concerns (adaptive capacity, reconstruction quality, comparison with Tripod) were directly addressed with new experiments and clearer explanations.
	•	Reviewer EAMz: Likely to remain 8 → 8
The reviewer’s main questions about adaptivity, trade-offs, and qualitative evidence were comprehensively addressed, reinforcing an already positive assessment.
	•	Reviewer Avba: Likely to move from 6 → 6
Novelty concerns are partially mitigated by clearer positioning and expanded related work, though some skepticism about empirical gains may remain.

---

### Decision · Program_Chairs · 2026-01-26

Accept (Poster)